# Auxin-induced signaling protein nanoclustering contributes to cell polarity formation

Xue Pan [1,2], Linjing Fang [3], Jianfeng Liu[2], Betul Senay-Aras[4], Wenwei Lin[1,2], Shuan Zheng[1], Tong Zhang[3], Jingzhe Guo[2], Uri Manor [3], Jaimie Van Norman [2], Weitao Chen [4✉] & Zhenbiao Yang [2✉]

Cell polarity is fundamental to the development of both eukaryotes and prokaryotes, yet the mechanisms behind its formation are not well understood. Here we found that, phytohormone auxin-induced, sterol-dependent nanoclustering of cell surface transmembrane receptor kinase 1 (TMK1) is critical for the formation of polarized domains at the plasma membrane (PM) during the morphogenesis of cotyledon pavement cells (PC) in *Arabidopsis*. Auxin-induced TMK1 nanoclustering stabilizes flotillin1-associated ordered nanodomains, which in turn promote the nanoclustering of ROP6 GTPase that acts downstream of TMK1 to regulate cortical microtubule organization. In turn, cortical microtubules further stabilize TMK1- and flotillin1-containing nanoclusters at the PM. Hence, we propose a new paradigm for polarity formation: A diffusive signal triggers cell polarization by promoting cell surface receptor-mediated nanoclustering of signaling components and cytoskeleton-mediated positive feedback that reinforces these nanodomains into polarized domains.

[1] FAFU-UCR Joint Center for Horticultural Biology and Metabolomics Center, Haixia Institute of Science and Technology, Fujian Agriculture and Forestry University, Fuzhou, Fujian, China. [2] Center for Plant Cell Biology, Institute of Integrative Genome Biology, and Department of Botany and Plant Sciences, University of California, Riverside, CA 92521, USA. [3] Waitt Advanced Biophotonics Center, The Salk Institute for Biological Studies, 10010 N Torrey Pines Road, La Jolla, CA 92037, USA. [4] Department of Mathematics, University of California, Riverside, CA 92521, USA. ✉email: weitaoc@ucr.edu; yang@ucr.edu

Cell polarity is a fundamental cellular property required for the specialization and function of essentially all cells. It is now widely accepted that the plasma membrane (PM) is highly compartmentalized into specialized functional domains with distinct protein and lipid compositions. The formation of cell polarity depends on breaking of the PM symmetry to form spatially segregated domains at the cell surface[1,2]. How the spatiotemporal organization of signaling components is initiated, especially by uniform signals, remains unclear, despite decades of studies on the mechanisms behind the establishment and maintenance of cell polarity. To date, two prevalent models have been proposed to explain self-organizing cellular symmetry breaking or that induced by spatial cues: asymmetric recruitment of polarity signaling proteins or endomembrane-linked trafficking of these proteins[3–6]. However, these models have difficulty in explaining how lateral segregation of signaling components into polarized membrane domains is achieved, especially by a uniform field of highly diffusible small signaling molecules. Furthermore, although the importance of lipid environment for the polarized sorting of proteins is well-established in yeast and animal epithelial cells[7,8], the mechanisms by which membrane lipids contribute to the formation of polarized domains remain poorly understood.

Unequivocal evidence shows that many cell surface-signaling proteins are not uniformly mixed with membrane lipids via free diffusion-based equilibrium, but laterally segregated into highly dynamic signaling nanoclusters with specific lipids on the PM[9–11]. Furthermore, the spatiotemporal distribution of membrane lipids is also known to play a critical role in regulating signaling events. Among these lipids, cholesterol and related sterols often appear organized in clusters and have the ability to form tightly ordered domains together with sphingolipids[12]. These domains, known as "membrane rafts", have long been suggested as a crucial platform for protein sorting[7] and signal transduction at the cell surface[12], particularly in the spatial regulation of key cellular processes, such as cell polarization[7,13] and immune response[14]. Notably, members of the Rho family GTPases, such as Ras, Cdc42, and ROP6 (a Rho-like GTPase from plants), appear to form transient nanoclusters on the PM. Anionic lipids, such as phosphatidylserine, have been shown to be key structural components of these nanoclusters and thus affect their signaling[15–17]. However, little is known about how distinct nanodomain-based signaling platforms are generated and maintained by specific signals and how they may contribute to the spatiotemporal regulation of cellular processes.

Auxin, a highly diffusible small-molecule hormone known to regulate cell polarity[18], impacts virtually every aspect of plant growth and development via nuclear gene transcription and rapid non-transcriptional responses controlled by the nuclear/cytoplasmic TIR1/AFB auxin receptors[19,20], as well as those mediated by the PM-localized transmembrane kinases (TMKs)[21,22]. Auxin-induced activation of ROPs is involved in cellular symmetry breaking such as polarization of PIN proteins, which export auxin[21,23], and the morphogenesis of *Arabidopsis thaliana* leaf and cotyledon pavement cells (PCs)[21,24,25]. These cells form the puzzle-piece shape with interlocking lobes and indentations, which require the establishment of multiple alternating polarized domains for the formation of lobes and indentations, respectively (Fig. 1). In PCs, the formation of these polarized domains requires the TMK-dependent activation of ROPs by auxin[25]. In particular, ROP6 is polarly localized to and defines the indentation-forming regions where it promotes the ordering of cortical microtubules (CMT)[26,27]. However, the mechanisms underlying the lateral segregation of signaling components, such as ROP6, into functional polarized domains at the PM during PC formation remain elusive. Inspired by the involvement of

membrane lipids in the formation of distinct nanodomains, we hypothesized that membrane lipids exert a similar function during auxin-induced polarity formation in PCs.

Here, we tested this hypothesis via genetic, biochemical, and cell biological approaches. On the basis of our findings, we propose a working model for auxin-triggered polarized domain formation at the PM via auxin-induced nanoclustering of sterol lipids and PM-associated signaling proteins. Auxin first induces the appearance of asymmetric distribution of signaling components by promoting the formation of interdependent TMK1 nanoclusters and ordered lipid nanodomains, and then this initial asymmetry is reinforced to form the polarized domains by microtubule-mediated diffusion restriction of these TMK1/ordered lipid nanodomains.

## Results

**Sterol-rich domains define indentations of pavement cells**. To assess the role of PM nanodomains in the morphogenesis of the puzzle-piece-shaped PCs, we first analyzed the distribution of ordered lipid domains in cotyledon PCs using di-4-ANEPPDHQ, a lipid order-sensitive probe[28,29]. Based on the fluorescent intensity of the probe in two different spectral channels, green (ordered) and red (disordered), a ratiometric general polarized image (GP image) was generated. High GP value (increased green fluorescence compared with red) represents high order of membrane lipids (Fig. 1a–e). Plasmolysis with 0.8 M mannitol showed that the peripheral staining was associated with the PM, but not with the cell wall (Supplementary Fig. 1). The optical resolution limit of confocal microscopy prevents the mid-plane distinction of complementary lobe and indenting regions of the PM in two adjacent PCs. As previously described[26,30], therefore, we used optical sections just above (1–1.5 μm) the cell mid-plane to separate the complementary PM regions of the adjacent cells. As shown in Fig. 1a–e, the GP images and quantitative analyses revealed that the localization of ordered lipid domains was biased toward the indenting region.

Sterols are known as a key component of ordered domains in the PM of yeast and mammalian cells[12,13]. We thus visualized the distribution of sterols in PCs by staining cotyledons with filipin, a sterol-binding probe commonly used for in situ sterol localization in various organisms, including plants[31]. PCs exhibited a patchy pattern of filipin–sterol complexes along the PM, which are preferentially distributed to indenting regions (Supplementary Fig. 2A–D and Supplementary Movie 1). To complement these staining methods, we also attempted fluorescent protein-tagged flotillin1 as a live molecular marker for sterol-rich lipid domains. In mammalian cells, flotillin has been shown to co-localize and associate with cholesterol-rich microdomains using both biochemical[32,33] and microscopy methods[34–36]. Flotillin is conserved in plants and found in PM nanodomains in Arabidopsis[37]. Arabidopsis flotillin1 is also enriched in detergent-resistant membrane (DRM) fractions[37], which likely contain membrane raft components, indicating its potential association with sterol-rich domains. To further test whether flotillin1 is a useful marker for sterol-rich nanodomains in plants, we generated a transgenic line expressing flotillin1-mVenus under its native promoter. As shown in Supplementary Movies 2 and 3, flotillin1 proteins form dynamic nanoclusters in the PM. These flotillin1 nanodomains strongly colocalized with the patched filipin–sterol complexes in the PM of PCs (Supplementary Fig. 2E–H), but not the general PM marker LIT6b (Supplementary Fig. 2I–L). These results suggest that flotillin1 is a conserved marker for sterol-rich nanodomains. Consistent with the staining results, in the majority of cases, flotillin1 proteins were more abundant in indentations than in lobes

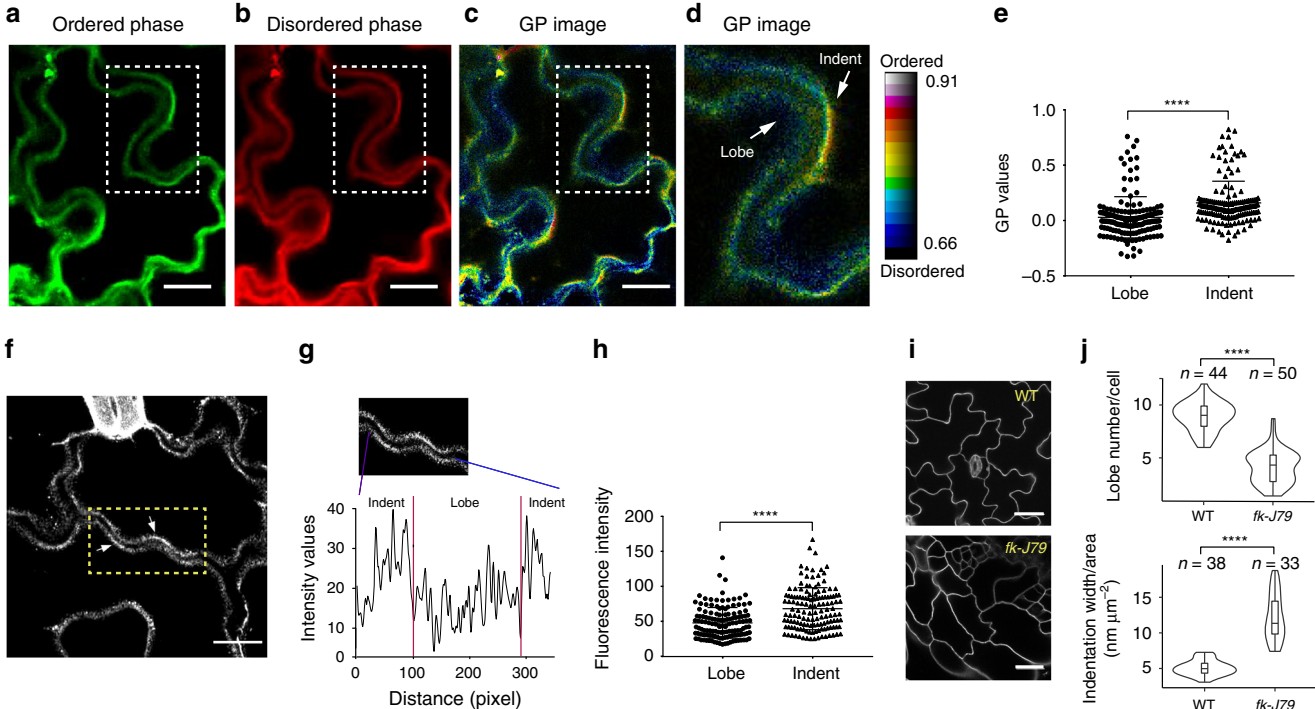

**Fig. 1 Ordered membrane domains are preferentially localized to indenting regions. a**, **e** Plasma membrane order visualization using di-4-ANEPPDHQ staining in the pavement cells of 2–3-day-old *Arabidopsis* cotyledons (Col-0). **a**–**d** Representative images obtained after di-4-ANEPPDHQ staining. **a** Di-4-ANEPPDHQ fluorescence recorded between 500 and 580 nm, representing high lipid ordering. **b** Di-4-ANEPPDHQ fluorescence recorded between 620 and 750 nm, representing low lipid ordering. **c** A radiometric color-coded GP image generated from **a** and **b**[28,29]. **d** An enlarged GP image corresponding to the boxed areas in **c**. The GP image is a false-color image, which runs over the range indicated by the color bar. Color bar values represent GP values ascend from bottom to top, with red colors indicating high membrane ordering, whereas blue colors indicating low membrane ordering. Scale bars = 15 μm. **e** Quantitative analysis of mean GP values obtained from the complementary lobing and indenting regions of 161 sites of 56 cells from three independent experiments. GP values at indenting regions are significantly higher than that at lobing regions. **f**–**h** Flotillin1-mVenus shows a polar distribution toward indenting regions. **f** Representative image showing the distribution of flotillin1-mVenus in PCs of 2–3-day-old cotyledons. The region highlighted in the dotted-line box is further analyzed in **g**. Scale bars = 15 μm. **g** Fluorescent intensity values scanned along the indicated region in **f**. **h** Quantitative analysis of fluorescence intensity at the complementary lobing and indenting regions of 138 sites of 45 cells from three independent experiments. **i**, **j** The sterol biosynthesis mutant *fackel-J79* (*fk-J79*), which is defective in gene encoding a sterol C14 reductase, shows altered PC shape. **i** Representative images of cotyledon pavement cells in *fk-J79* mutant and its corresponding wild type. Scale bars = 30 μm. **j** Quantitative analysis of the number of lobes and indentation widths in PCs of *fk-J79* mutant and its wild type. *n* represents the number of cells. Data are representative of three independent experiments which have the same pattern. ****$P \leq 0.0001$.

(Fig. 1f–h; Supplementary Movie 4). Collectively, these results show that ordered lipid domains are preferentially distributed to indenting regions of the PM, as is the active ROP6 GTPase[26].

To evaluate the roles of sterols in PC morphogenesis, we examined PC phenotypes of sterol biosynthesis mutants (*fk-J79*, *hyd1-E508*, *cpi1-1*) with an overall reduction in the major sterols (sitosterol and stigmasterol) found in the PM[38–40]. All these mutants display greatly altered PC shapes with reduced lobe numbers and increased indentation width compared with their wild-type control (Fig. 1i, j; Supplementary Fig. 3A, B). Importantly, the depletion of sterol from the PM by methyl-β-cyclodextrin (mβCD)[41,42], a sterol-depleting agent, also caused a similar defect in PC morphogenesis and compromised auxin-promoted PC interdigitation (Supplementary Fig. 3C, D). These defects resemble those induced by the disruption of the ROP6 pathway that promotes the organization of cortical microtubules and the formation of PC indentation[24,26]. Because sterol perturbation either by genetic or pharmacological approaches may generate pleiotropic effects apart from the disruption of sterol-rich nanodomains, one cannot exclude the possibility that PC morphogenesis is not directly linked to these nanodomains. However, the similar PC shape changes (i.e., increased indentation widths and reduced lobe numbers) resulting from defects in

ROP6 signaling and sterol accumulation, together with the preferential distribution of both ROP6 and sterol-rich ordered lipid nanodomains to the indenting regions, strongly support a role for the sterol nanodomains in defining and establishing the indenting regions, herein termed the indentation polarity.

**Auxin induces lipid ordering required for ROP6 activation.** Our previous studies show that auxin promotes the formation of the puzzle-piece PC shape, suggesting that auxin is a signal that initiates the formation of the PC polarity[21,25]. Thus, we asked whether auxin regulates the generation of ordered lipid domains in PCs. The auxin biosynthesis mutant *wei8-1tar2-1* with reduced auxin levels in cotyledons[43] exhibited greatly reduced lipid ordering in PCs, as indicated by di-4-ANEPPDHQ staining (Fig. 2a, b). This defect in lipid ordering was rescued by exogenous auxin (Fig. 2a, b). Similar to the *wei8-1tar2-1* mutant, the sterol biosynthesis mutant *fk-J79* also exhibited reduced lipid ordering (Fig. 2c, d). However, unlike the *wei8-1tar2-1* mutant, the *fk-J79* mutant was completely insensitive to exogenous auxin in the promotion of lipid ordering (Fig. 2a–d). Furthermore, the auxin-induced increase in the number of polar sites (as indicated by the lobe number) in PCs was completely abolished in the *fk-J79* mutant (Fig. 2e, f).

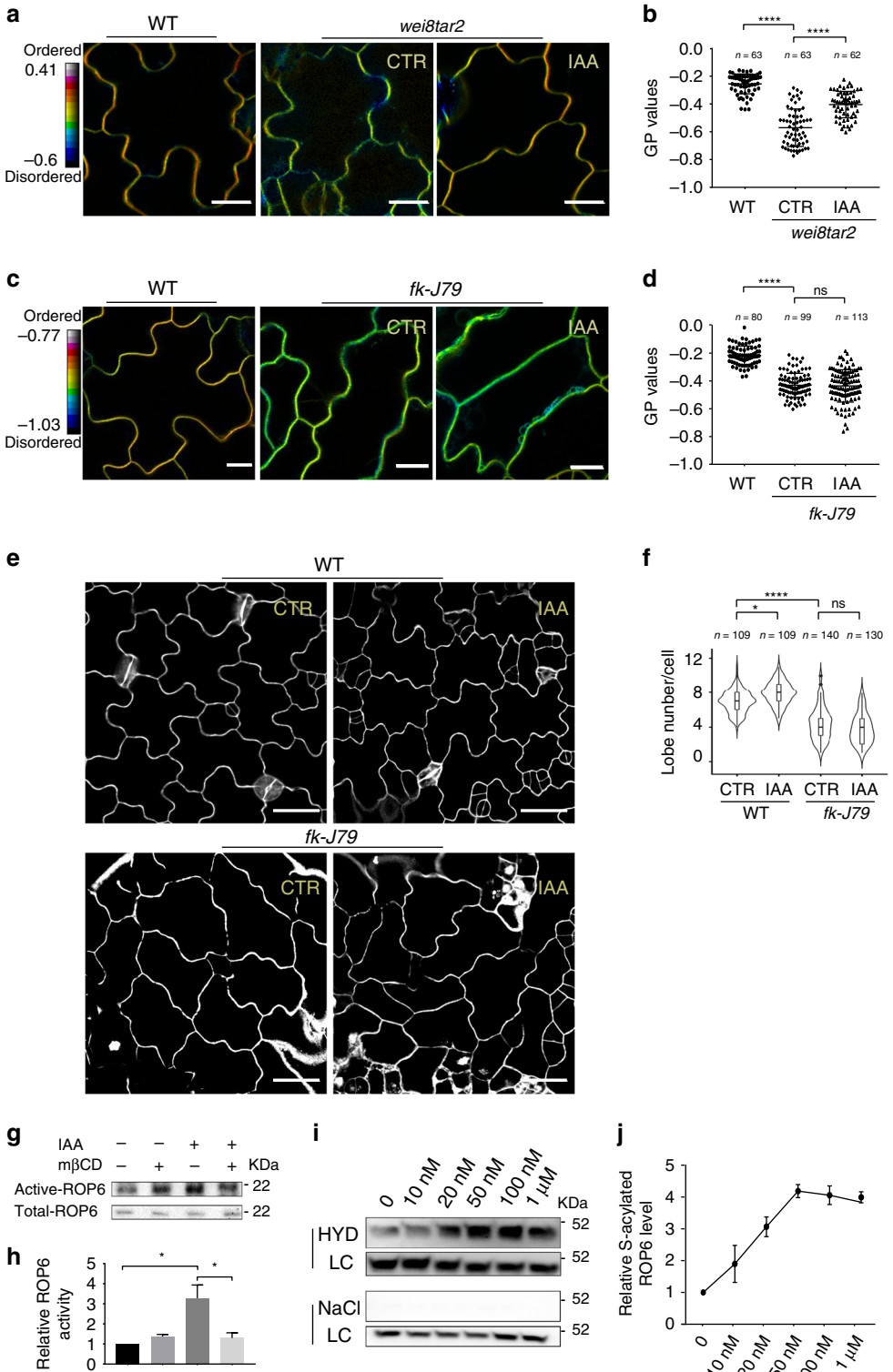

To understand how ordered lipid domains contribute to the formation of the indentation polarity, we examined the effect of sterols on the activity of ROP6, an indentation polarity marker critical for indentation formation[21,26,27]. Previously we showed that the activity of ROP6 is promoted by auxin[21,25]. We further found that reducing PM sterols by mβCD treatment greatly hindered auxin-induced increase in ROP6 activity (Fig. 2g, h), suggesting that ordered lipid domains are important for the auxin promotion of ROP6 activity. Previous biochemical evidence showed that S-acylation contributes to the recruitment of ROP6 into lipid-raft like, detergent-resistant membrane fraction[44]. We further showed that auxin increased the amount of S-acylated ROP6 in a dose-dependent manner (Fig. 2i, j), consistent with the notion that auxin promotes the association of ROP6 with sterol-rich ordered lipid domains. Therefore, we propose that auxin promotes the formation of ordered lipid domains required for the ROP6-dependent indentation formation.

**Fig. 2 Auxin promotes plasma membrane (PM) ordering required for ROP6 activation. a–d** The reduced PM ordering was rescued by auxin in the auxin biosynthesis mutant *wei8-1tar2-1*, but not in the sterol biosynthesis mutant *fk-J79*. **a, c** Representative GP images of pavement cells (PCs) in different genetic backgrounds with or without IAA treatments obtained after di-4-ANEPPDHQ staining. Note: To quantify the effect of auxin on the overall PM ordering, the cells were imaged at the mid-plane without separating complementary lobes and indentations. **b, d** Quantitative analysis of mean GP values extracted from the PM of multiple PCs in two independent experiments. **e, f** Auxin-promoted PC interdigitation was abolished in the *fk-J79* mutant. **e** Representative confocal images of PCs in wild-type and *fk-J79* mutant with or without the IAA treatment. **f** Quantitative analysis of the number of lobes in *fk-J79* mutant and its wild type with or without the IAA treatment. Data are representative of three independent experiments with the same pattern. **g** Representative western blot images of ROP6 activity assays. mβCD, methyl-β-cyclodextrin. **h** Quantitative analysis of the relative active ROP6 level from three independent experiments. The relative active ROP6 level was determined as the amount of GTP-bound ROP6 divided by the total amount of ROP6. **i** Representative western blot images of *S*-acylation assays. For each assay, the sample was compared with the loading control (LC) with or without hydroxylamine (HYD) for hydroxylamine-dependent capture of *S*-acylated proteins. The samples treated with NaCl instead of HYD was used as the negative control. **j** Quantitative analysis of the relative *S*-acylated ROP6 level from three independent experiments. The relative *S*-acylated ROP6 level was determined as the amount of *S*-acylated ROP6 divided by the amount of total GFP-ROP6. The relative active ROP6 level **h** or *S*-acylated ROP6 level **j** in different treatments was normalized to that from the mock-treated control as standard 1. Data are presented as mean ± SD. *n* in **b, d, f** represents the number of independent cells. WT, wild type. CTR, control. Scale bars: 15 μm **a** and **c**, and 30 μm **e**. *$P < 0.05$, ****$P \leq 0.0001$; ns, not significant.

**Auxin promotes the formation of TMK1 nanoclusters.** We next investigated how auxin promotes the formation and polarization of ordered lipid domains to the indenting region. Auxin signaling is mediated by at least two major pathways: the nuclear-based TIR1/AFB pathway that regulates the expression of auxin-induced genes[19,20] and the PM-based TMK pathway involved in both transcriptional and non-transcriptional auxin responses[21,22]. Because auxin promotes ROP6 activation in a TMK-dependent manner[21,25], we speculated that the TMK-dependent pathway regulates lipid ordering and its polarity. In addition, the *tmk1tmk4* double mutant showed reduced lipid ordering in the PM of PCs and was insensitive to the auxin-induced lipid ordering (Supplementary Fig. 4). As an initial step in testing our hypothesis, we analyzed the spatiotemporal dynamics of TMK1, a major member of the TMK clade with four functionally overlapping members[25]. To study dynamics of TMK1, time-lapse images were collected on a Zeiss microscopy equipped with a total internal reflection fluorescence (TIRF) module for 2–3-day-old cotyledon PCs followed by single-particle-tracking method, which is commonly used to visualize the dynamics of PM-localized nanoscale compartments including protein nanoclusters[45–47]. As illustrated in Fig. 3a, anticlinal surfaces of PCs form interlocking lobes and indents, but it is technically impossible to image these surfaces using TIRF or other single-molecule imaging methods. Therefore, we investigated how auxin affects the dynamic behaviors of nanoparticles on the periclinal surface of PCs. The results showed that TMK1-GFP, expressed under *TMK1*'s native promoter, was laterally segregated as nanoparticles at the PM (Supplementary Movie 5 and Fig. 3b). Treatment with 100 nM IAA resulted in a dramatic decrease in the density of TMK1-GFP particles (Fig. 3c), accompanied with an increase in the particle size (Fig. 3d, e). These results suggest that auxin promotes the formation of larger TMK1 nanoclusters, possibly by coalescing single proteins or small and unstable nanoclusters. Further diffusion analyses showed that a two-component model describes the distribution of diffusion coefficients much better than a one-component model (Supplementary Fig. 5A), indicating TMK1-GFP particles on the membrane consist of at least two populations, one with slow and one with fast diffusion. IAA treatment (100 nM, 10 min) resulted in a significant reduction in diffusion coefficients of both populations (Fig. 3f; Supplementary Table 1). These results suggest that auxin specifically promotes the formation of the TMK1 nanoclusters with larger size and slower diffusion.

**TMK1 nanoclustering and lipid ordering are interdependent.** Because ligand-induced receptor clustering may modulate the local lipid environment and stabilize ordered lipid domains around the receptor[48,49], we hypothesize that auxin-triggered TMK1 nanoclustering stabilizes a local ordered lipid environment. To test the hypothesis, we first studied the dynamic relationship between TMK1 and flotillin1 proteins at the PM. As shown in Fig. 3g–i, TMK1-tagRFP particles are partially colocalized with flotillin1-mVenus particles. As a protein associated with ordered lipid domains, flotillin1 is expected to be present in diverse populations of nanoclusters, and thus its partial colocalization with TMK1 is anticipated. To functionally test the role of ordered lipid domains in TMK1 nanoclustering, we analyzed TMK1 dynamics in the presence of the sterol-disrupting agent mβCD (10 mM for 30 min). This treatment greatly hindered the auxin-induced increase in sizes (Fig. 3e) and reduction in diffusion of TMK1 particles (Fig. 3f; Supplementary Table 1). These results indicate that auxin-induced TMK1 nanoclustering is dependent upon sterol content in the PM.

To test whether auxin-induced changes in lipid ordering in the PM requires TMKs, we analyzed the dynamics of flotillin1-mVenus under its native promoter, which was used as an indicator for the dynamics of ordered nanodomains at the PM (Supplementary Movies 2 and 3). Similar to TMK1, IAA treatment (100 nM, 10 mins) increased the flotillin1-mVenus particle size (Fig. 4a–d). Diffusion analyses also revealed the presence of a fast-diffusing population and a slow-diffusing population (Supplementary Fig. 5B). Auxin treatment increased the fraction of the slow-diffusing population with a decrease in the fraction of the fast-diffusing population (Figs. 4e, f; Supplementary Table 1). The diffusion coefficient of flotillin1-mVenus in *tmk1tmk4* double mutant was significantly faster than that in wild type (Fig. 4f). In addition, the *tmk1/tmk4* double mutant displayed a substantial resistance to the promotion of flotillin1 particle size (Fig. 4c, d) and the reduction of particle diffusion (Fig. 4f) triggered by auxin. These data demonstrate that a significant portion of auxin's effect on the dynamics of flotillin-associated ordered nanodomains is regulated via the TMK1-mediated pathway.

**TMKs are required for auxin-mediated nanoclustering of ROP6.** TMKs are required for auxin-induced changes in ordered lipid nanodomains (Fig. 4) and increases in ROP6 activity[21,25], while auxin-induced lipid ordering is critical for auxin-induced increase in ROP6 activity (Fig. 2). Thus, we hypothesize that auxin promotes the nanoclustering of active ROP6 in a TMK- and sterol-dependent manner. Indeed, single-particle-tracking analysis showed that the constitutively active (CA) form of ROP6 (YFP-CArop6$^{Q64L}$) was preferentially distributed in the particles with larger size and slower diffusion properties in comparison with the inactive form of ROP6 (dominant negative,

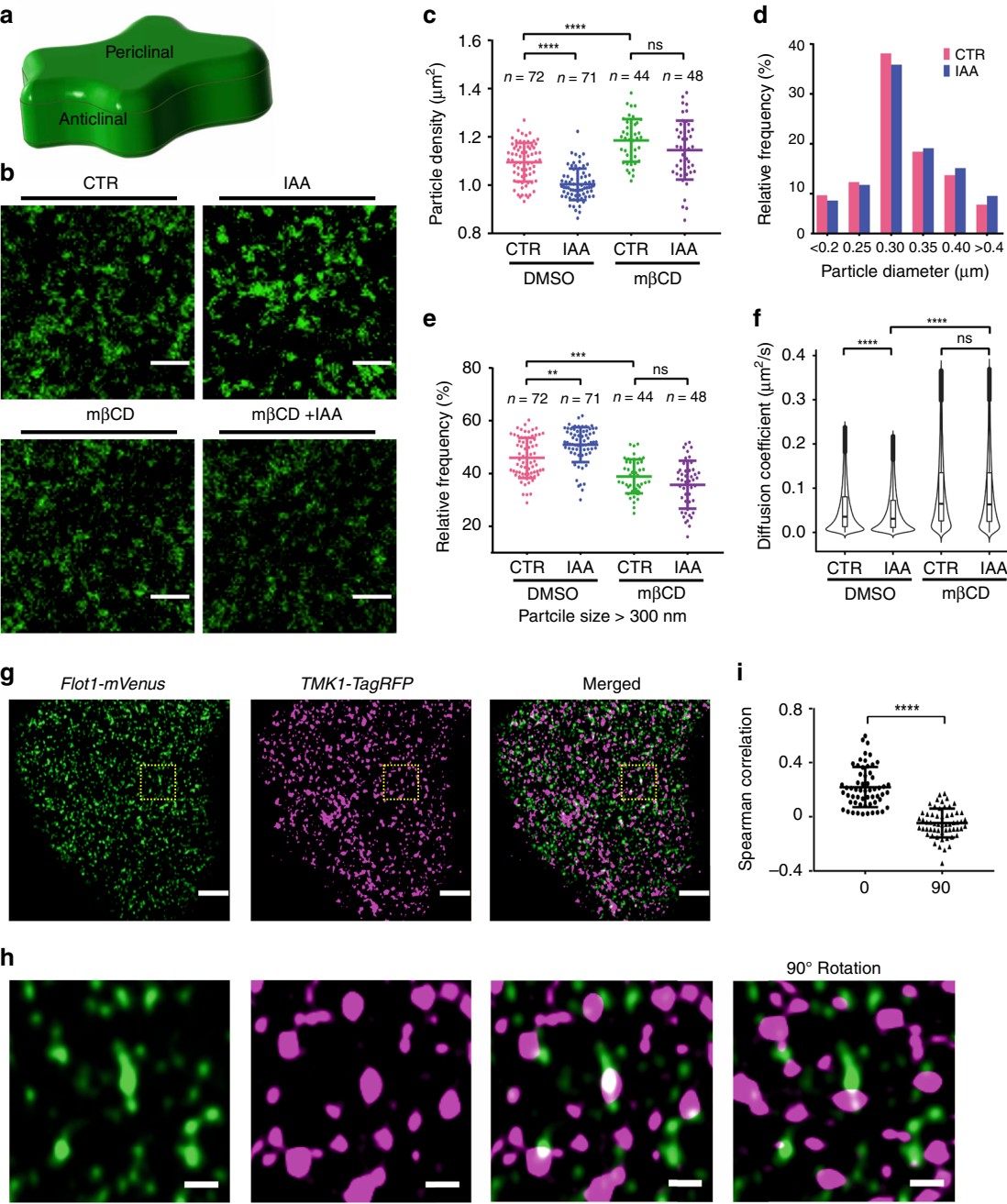

**Fig. 3 Auxin promotes TMK1 nanoclustering in a sterol-dependent manner. a** The 3D view of the pavement cell (PC). **b** Representative TIRF images of PCs expressing TMK1::TMK1-GFP with or without the IAA (100 nM, 10 min) treatment and the methyl-β-cyclodextrin (mβCD, 10 mM, 30 min) pretreatment. **c** Quantitative comparison of the TMK1-particle density in different treatments. **d** Histogram of TMK1-particle size distribution in control and IAA-treated cells. **e** Relative frequency of TMK1 particles with size larger than 300 nm. Data in **c** and **e** are presented as mean ± SD, and *n* represents the number of independent cells. **f** Comparison of diffusion coefficients of TMK1 particles in different treatments. The violin plot shows the distribution of the original diffusion coefficients without the log transformation. The distribution of log-transformed diffusion coefficients is shown in Supplementary Fig. 5. The number of particles analyzed in each treatment (from left to right) are 156,480, 131,187, 98,267, and 107,269, respectively. Statistical significance of treatment effects was evaluated on the overall diffusion coefficient of each treatment. CTR, control. Data in **c**, **d**, **e**, and **f**) are representative of three independent experiments with the same pattern. **g–i** Colocalization analysis of PCs co-expressing flotillin1::flotillin1-mVenus and TMK1::TMK1-tagRFP. **g** Representative Airyscan images. The yellow dashed boxes indicate the regions depicted in **h**. To test the specific colocalization, the merged image was generated by rotating the image from the red channel with respect to the image from the green channel by 90°, a condition in which only random colocalization can be observed[78]. **i** A statistical analysis comparing Spearman's rank-correlation coefficients in nonrotated (0°) and 90° rotated images. The significantly higher correlation coefficients obtained in nonrotated images than in 90° rotated images suggests that the observed colocalization is not due to random chance. Data were obtained from 58 regions of 20 cells in two independent experiments. Scale bars: 2 μm **b**, 3 μm **g**, and 0.5 μm **h**. **P ≤ 0.01, ***P ≤ 0.001, ****P ≤ 0.0001; ns, not significant.

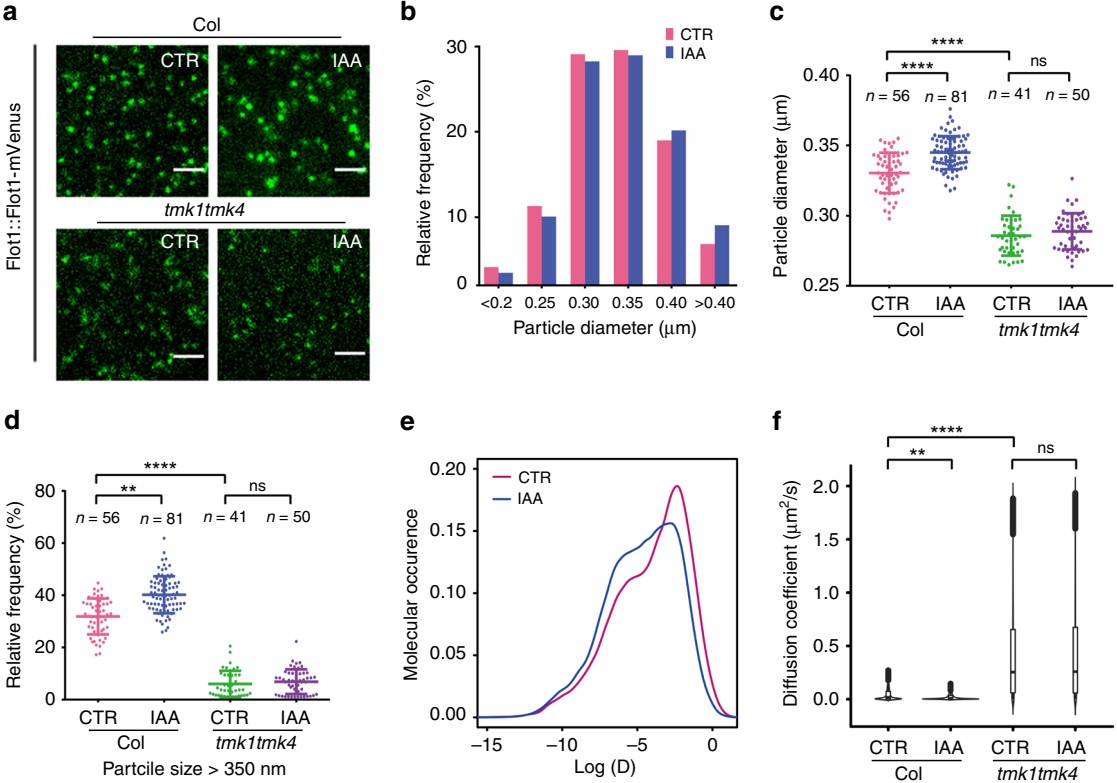

**Fig. 4 TMK contributes to the auxin-promoted nanoclustering and stabilization of flotillin1 particles. a** Representative TIRF images of pavement cells expressing flotillin1:: flotillin1-mVenus with or without IAA (100 nM, 10 min) treatment in wild type *(Col-0)* and *tmk1tmk4* double-mutant background. Scale bars = 2 μm. **b** Histogram of flotillin1 particle size distribution in control and IAA-treated cells. **c** Quantitative comparison of flotillin1 particle diameters in different treatments. **d** Relative frequency of flotillin1 particles with size larger than 350 nm. Data are presented as mean ± SD. *n* represents the number of independent cells. **e** The density curves of diffusion coefficients of flotillin1 particles in control (pink curve) and IAA-treated (blue curve) cells. **f** Comparison of diffusion coefficients of flotillin1 particles in different treatments. The number of particles analyzed in each treatment (from left to right) are 49,216, 43,223, 47,078, and 48,206, respectively. CTR, control. Data in **b**–**f** are representative of three independent experiments with the same pattern. **P ≤ 0.01, ****P ≤ 0.0001; ns, not significant.

*DNrop6*$^{D121A}$) (Fig. 5a–c; Supplementary Fig. 5D and Supplementary Table 1). The *DNrop6*$^{D121A}$ line is different from the line (*DNrop6*$^{T30N}$) used by Poraty-Gavra et al.[50]. Unlike *DNrop6*$^{T30N}$, the recruitment of DN*rop6*$^{D121A}$ to the PM is not compromised and does not accumulate in nuclei (Supplementary Fig. 6). To test the effect of auxin on ROP6 nanoclustering, we performed single-particle tracking of mEGFP-ROP6 in a *ROP6p::mEGFP-ROP6* line. mEGFP-ROP6 showed clear lateral segregation on the PM (Fig. 5d; Supplementary Movie 6). Treatment with 100 nM IAA, which induces ROP6 activation by fourfolds[21], significantly increased ROP6 particle sizes (Figs. 5d, e). Further analyses showed that, like TMK1 particles, the distribution of diffusion coefficients of ROP6 particles was better described by a two-component model, indicating the existence of at least two populations of ROP6 particles, one slow- and one fast-diffusing populations (Supplementary Fig. 5C). Consistently, IAA treatment significantly reduced the ROP6 diffusion coefficient with an increase in the fraction of the slow-diffusion population (Fig. 5f; Supplementary Fig. 5E and Supplementary Table 1). In addition, the effect of auxin on the size increase and diffusion decrease for ROP6 particles was completely abolished in the presence of sterol-depleting agent mβCD (Fig. 5g, h). Furthermore, ROP6 and flotillin1 particles partially reside in the same nanodomains upon the auxin treatment (Supplementary Fig. 7). Together, these data suggest that auxin-triggered ROP6 activation induces ROP6 stabilization in sterol-rich nanodomains with increased size and reduced diffusion. Similar auxin-induced formation of ROP6

nanoclusters were recently observed in roots by both photo-activated localization microscopy and TIRF imaging[16].

We next asked whether TMK1/4 also regulates the formation of ROP6 nanoclusters. Indeed, in the *tmk1tmk4* double mutant, the ROP6 particles are much smaller in size (Fig. 5d, e) and have a higher mobility (Fig. 5f). In addition, the auxin-triggered ROP6 particle size increase (Fig. 5d, e) and diffusion decrease (Fig. 5f) were greatly compromised in the *tmk1tmk4* double mutant, indicating that TMK1/4 are critical for auxin-induced changes in the dynamic behavior of ROP6 particles. Taken together, these data strongly support the hypothesis that TMK1/4 regulates auxin-induced ROP6 nanoclustering via shaping the sterol-rich lipid environment at the PM. Interestingly, Platre et al.[16] showed that phosphatidylserine lipid is also important for auxin-mediated ROP6 nanoclustering in roots, and phosphatidylserine biosynthesis mutants exhibit severe PC morphogenesis defect. Thus, ROP6 nanoclusters in PCs are likely enriched in both sterols and phosphatidylserine.

**CMTs stabilize TMK1/ordered lipid nanodomains.** Our data described above show that auxin induces both the formation of the indentation polarity and sterol-dependent nanoclustering of TMK1 and ROP6, which promote cortical microtubule (CMT) organization during PC development[24–26]. Interestingly, we found that a portion of constitutively active (CA) ROP6 particles was localized along CMTs (Supplementary Fig. 8A–C), suggesting

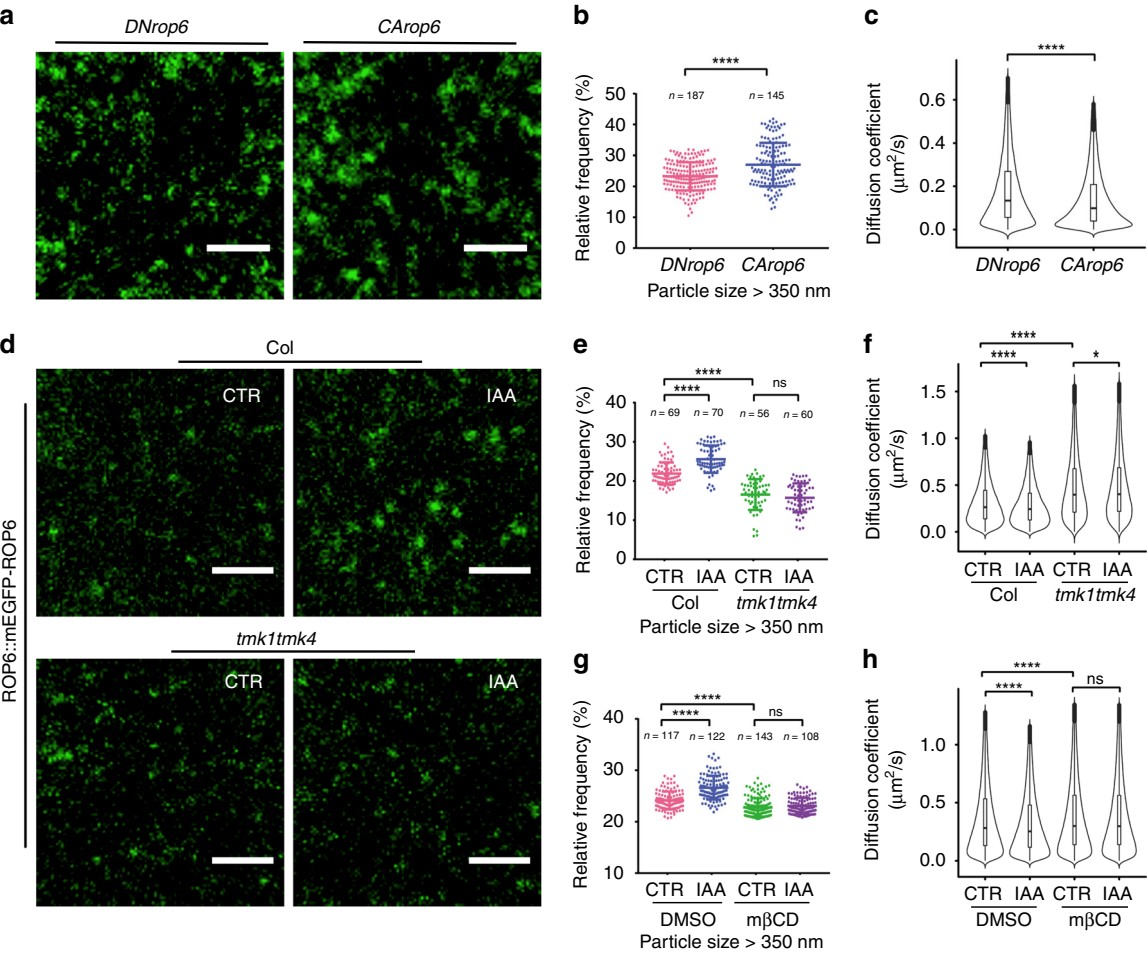

**Fig. 5 TMK contributes to the auxin-promoted nanoclustering and stabilization of ROP6 particles. a–c** *CArop6* formed larger particles with slower diffusion rate than *DNrop6* at the plasma membrane. **a** Representative TIRF images of pavement cells (PCs) expressing 35S::YFP-*CArop6* and 35S::YFP-*DNrop6*. **b** Relative frequency of YFP-*CArop6* and YFP-*DNrop6* particles with size larger than 350 nm. **c** Diffusion coefficients of YFP-*CArop6* and YFP-*DNrop6* particles. The number of particles analyzed for YFP-*DNrop6* and YFP-*CArop6* lines are 97,390 and 86,712, respectively. **d–f** Auxin-induced increase in size and decrease in diffusion rate of ROP6 particles was compromised in the *tmk1tmk4* double mutant. **d** Representative TIRF images of PCs expressing ROP6::mEGFP-ROP6 in wild type *(Col-0)* and *tmk1tmk4* mutant treated with or without IAA (100 nM, 10 min). **e** Relative frequency of ROP6 particles with size larger than 350 nm. **f** Comparison of diffusion coefficients of ROP6 particles in different treatments. The number of particles analyzed in each treatment (from left to right) are 40,685, 42,016, 32,663, and 32,893, respectively. **g, h** Auxin-induced increase in size and decrease in diffusion rate of ROP6 particles was abolished in the presence of the sterol-depleting agent methyl-β-cyclodextrin (mβCD). **g** Relative frequency of ROP6 particles with size larger than 350 nm in control and IAA-treated cells with or without mβCD pretreatment. **h** Comparison of diffusion coefficients of ROP6 particles in different treatments. The number of particles analyzed in each treatment (from left to right) are 354,303, 355,561, 458,949, and 411,675, respectively. CTR, control. Data in **b**, **e**, **g** are presented as mean ± SD, and *n* represents the number of independent cells. All data are representative of three independent experiments with the same pattern. Scale bars = 2 μm. *$P < 0.05$, ****$P \leq 0.0001$; ns, not significant.

a potential physical connection between CMTs and active ROP6s. Thus, we hypothesize that CMTs may stabilize TMK1 nanoclusters and ordered lipid nanodomains at the PM. CMTs are physically attached to the inner leaflet of the PM, and thus could presumably hinder the mobility of TMK1 nanoclusters and ordered lipid nanodomains. To investigate whether CMTs regulate the TMK1 nanoclusters, we observed an immediate effect of microtubule disruption on TMK1 nanoclusters by treating the cotyledons expressing TMK1-GFP or flotillin1-mVenus with 5 μM oryzalin for 30 min. The effectiveness of the oryzalin treatment in disrupting CMTs was demonstrated by treating cotyledons expressing the *35 S::GFP-TUB* construct with the corresponding drug (Supplementary Fig. 8D, E). As shown in Fig. 6a–c, CMT disruption with oryzalin greatly decreased the size and increased the diffusion of TMK1 particles on the PC. Flotillin1 particles became highly diffusive upon oryzalin treatment, even though no significant changes in particle size were observed

(Fig. 6d–f). In addition, TMK1 and flotillin1 particles displayed largest particle sizes and slowest diffusion rates in the presence of auxin and intact CMTs, and depolymerizing CMTs did not completely block the auxin-induced diffusion decrease (Fig. 6a–f), suggesting that the stabilization of TMK1 and flotillin1 particles is regulated by auxin-induced nanoclustering and the subsequent CMT-mediated diffusion restriction. To complement the pharmacological approach, we further studied the dynamics of TMK1 and flotillin1 particles in the *ktn1* mutant, which has altered microtubule organization and PC shape[27]. As shown in Supplementary Fig. 9A–D, the particles of TMK1 and flotillin1 displayed smaller size and faster diffusion rate in the *ktn1* mutant than in the wild type. In addition, the dynamics of TMK1 particles were studied in the temperature-sensitive *mor1-1* mutant, in which CMTs become shortened and disorganized when grown at the restrictive temperature (29 °C) for 2 h[51]. The diffusion rate of TMK1 particles increased in both wild-type and *mor1-1* mutant

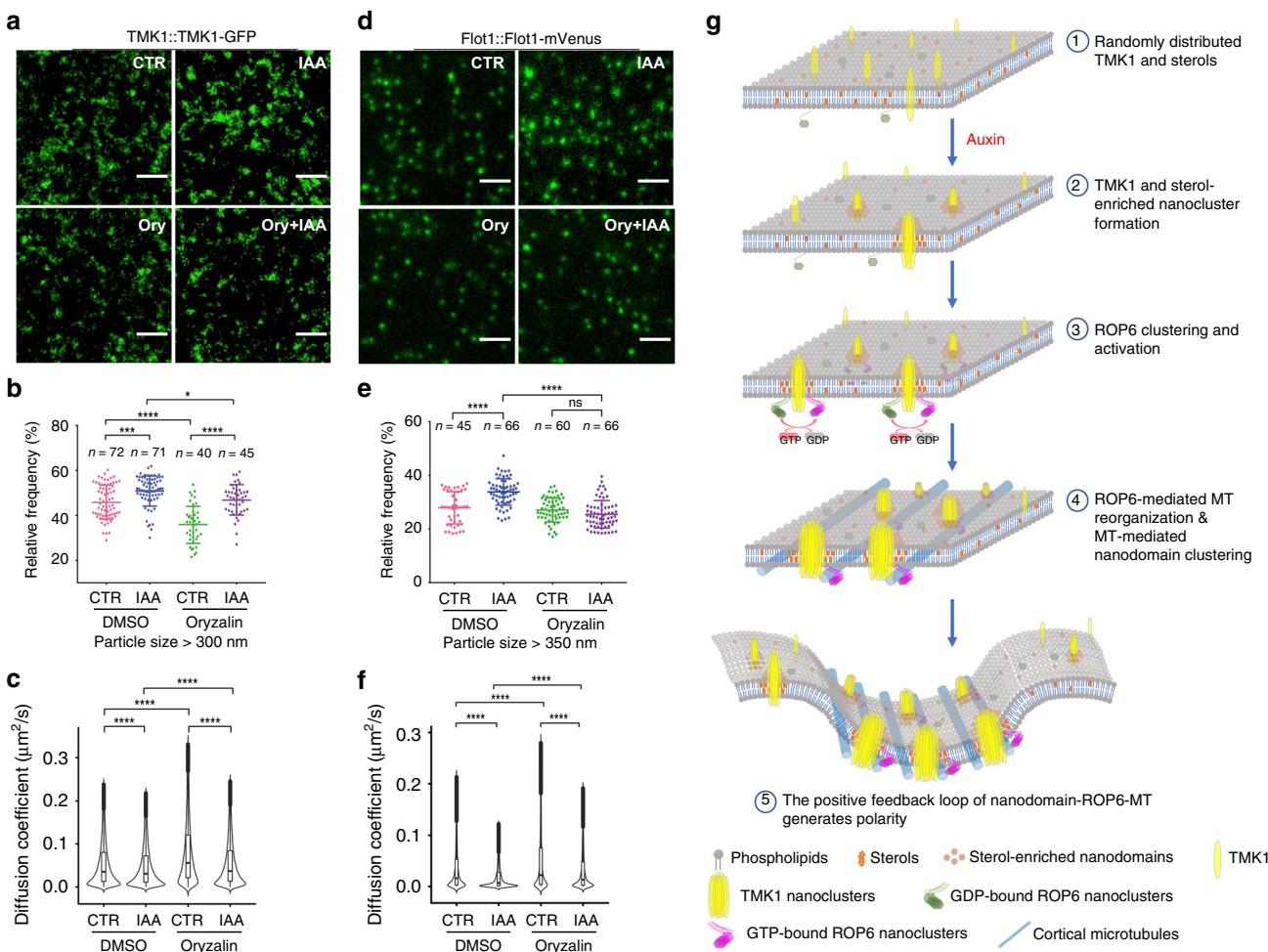

**Fig. 6 Cortical microtubules stabilize TMK1 and flotillin1 nanoclusters at the plasma membrane. a–c** Auxin-induced dynamic changes of TMK1 particles were not fully compromised by microtubule disruption. **a** Representative TIRF images of pavement cells (PCs) expressing TMK1::TMK1-GFP with or without IAA (100 nM, 10 min) and oryzalin (5 μM, 30 min) treatments. **b** Relative frequency of TMK1 particles with size larger than 300 nm. **c** Comparison of diffusion coefficients of TMK1 particles in different treatments. The number of particles analyzed in each treatment (from left to right) are 156,480, 131,187, 85,436, and 96,457, respectively. Note: The oryzalin and mβCD treatments (Fig. 3) on TMK1 dynamics were performed in the same batch, and the same mock treatment was used. The data and representative images for the control group are the same for Fig. 3b–f and **a–c**. **d–f** Auxin-induced dynamic changes of flotillin1(Flot1) particles were not completely compromised by microtubule disruption. **d** Representative TIRF images of PCs expressing flotillin1:: flotillin1-mVenus with or without IAA (100 nM, 10 min) and oryzalin (5 μM, 30 min) treatments. **e** Relative frequency of flotillin1 particles with size larger than 350 nm. **f** Comparison of diffusion coefficients of flotillin1 particles in different treatments. The number of particles analyzed in each treatment (from left to right) are 36,937, 47,316, 53,899, and 50,214, respectively. CTR, control. **g** Schematic illustration showing the steps involved in the auxin-mediated polarity formation. Proteins and structures are not to scale. Data in **b** and **e** are presented as mean ± SD, and *n* represents the number of independent cells. Data in **b**, **c**, **e**, and **f** are representative of three independent experiments with the same pattern. Scale bars = 2 μm. *P < 0.05, ***P ≤ 0.001, ****P ≤ 0.0001; ns, not significant.

after treatment with the restrictive temperature for 2 h. However, the relative change of diffusion rate was much higher in the *mor1-1* mutant than in the wild type, and the temperature-triggered decrease in particle sizes was only significant in the *mor1-1* mutant (Supplementary Fig. 9E–G). Moreover, introducing the *CArop6* into the TMK1-GFP background increased the size and reduced the diffusion rate of TMK1 particles (Supplementary Fig. 9H, I). Importantly, the effect of *CArop6* expression was compromised by oryzalin treatments, indicating that intact CMTs are required for the *CArop6*-induced dynamic changes of TMK1 particles. These results demonstrate a critical role for ROP6-dependent CMTs in stabilizing TMK1 nanoclusters.

Taken together, we propose the following model for the formation of the PC polarity (Fig. 6g): (1) Upon auxin signaling, ordered lipid nanodomains and TMK1 proteins are clustered. TMK1 nanoclustering further promotes the formation of larger,

stabilized ordered lipid nanodomains, probably by coalescing small, unstable lipid nanodomains. (2) The newly formed large lipid nanodomains provide an ideal lipid environment for ROP6 nanoclustering and activation. (3) The active ROP6 interacts with downstream effectors and promotes CMT ordering, which further restrict the diffusion of TMK1 nanoclusters and their surrounding ordered lipid nanodomains. (4) The initial asymmetric distribution of active ROP6 is further amplified by the positive feedback loop of ordered lipid nanodomain-ROP6 signaling-CMT ordering, and eventually leads to the formation of the multiple alternating polarized ROP6 domains prior to the indentation formation.

## Discussion

Our findings reveal a nanodomain-centric principle for the formation of cell polarity and for auxin signaling at the PM. These

findings demonstrate a pivotal role of nanoclustering in controlling cell polarity formation, expanding its emerging roles in the regulation of fundamental processes, including cell spreading and endocytosis[52]. Importantly based on our findings, we proposed a new paradigm for cell polarity formation that can explain cell polarization that is self-organizing or induced by uniform or localized signals. Local protein recruitment and/or endomembrane trafficking have been prevailing models to explain cell polarization[3–6,53,54]. Here, we show that the coordination of nanoclustering of cell surface-signaling components with microtubule-dependent feedback regulation of nanocluster diffusion restriction is essential for auxin-mediated PC formation. This new paradigm likely provides a common design principle underlying polarity formation in different cell systems.

A number of cell polarity proteins such as Rac1 and Cdc42 are found in nanoclusters[15,55], but it is unclear whether their nanoclustering is functionally linked to cell polarity formation. ROP6 acts downstream of TMKs and promotes the organization of ordered CMTs, which in turn slow the diffusive movement of TMK1 and flotillin1 particles at the PM. This feedback loop-dependent regulation of nanocluster is distinct from the TMK-based ROP6 nanoclustering directly triggered by auxin and is essential for the PC formation. The CMT regulation of ROP6 nanoclustering appears to be at least partially mediated by the direct association of active ROP6 (in the nanoclusters) with CMTs, as we found a portion of active ROP6 is associated with CMTs as nanoclusters. This differs from the regulatory mechanism of flotillin1 particles imposed by CMTs as the majority of the flotillin1 particles do not associate with CMTs, but rather localize close to or within CMT corrals (Supplementary Fig. 8F–K). It is possible that sterol-rich nanodomains provide the microenvironment for converting inactive ROP6 to active ROP6, which then interacts with its downstream effectors and promotes the CMT ordering. The association of ROP6 with CMTs may make them leave sterol-rich nanodomains, presumably because of steric-hindrance effects between CMTs and sterol-rich nanodomains. This could also explain the observed partial colocalization of flotillin1 and ROP6 nanoclusters (Supplementary Fig. 7). A critical role for microtubules in cell polarization is well recognized in eukaryotic cells[56,57], and their association with Rho GTPase signaling proteins (ICR1, MDD1, RhoGEFs, etc.) is widely documented[26,58–60]. CMTs have been implicated in regulating PIN protein polarization, and ROP6 nanoclustering may also impact PIN polarity[16,61]. During symmetry breaking at starvation exit in fission yeast cells, microtubules are important for the formation of polarized sterol-rich membrane domains[13]. Previous studies have also implicated CMTs in restricting the diffusion of PM-associated signaling molecules, thus, promoting their clustering during signaling[62,63]. All these observations potentially link cytoskeletal modulation of signaling protein nanoclusters to cell polarity formation in various systems. Therefore, future studies should determine whether the coordination of ligand-activated nanoclustering and cytoskeleton-based diffusion restriction of nanoclusters serves as a general principle behind the formation of cell polarity. In addition, considering the impact of cell walls on the dynamics of PM-localized protein nanoclusters[47] and PC morphogenesis[64,65] revealed by recent studies, our future work will focus on how the interplay between cell wall components, PM nanodomains and the cytoskeleton regulates PC formation.

Emerging evidence suggests that ligand-induced formation of protein-lipid nanodomains at the PM serves as pivotal signaling platforms throughout eukaryotic cells. Here, we show that auxin-induced formation of TMK1 nanoclusters and ordered lipid nanodomains are interdependent. We propose that this ligand-mediated self-organizing lipid–protein interaction provides an important mechanism to generate a nanodomain signaling platform. This is in line with the ligand-induced transmembrane receptor sculpting of the surrounding ordered lipid environment proposed for animal cells[48,49,66]. Such clustering of ordered lipid-associated proteins modulates the activity of signaling proteins associated with the inner leaflet of the PM, such as ROP6 GTPase as we showed here and other GTPases in animal systems[67,68]. Notably, auxin-induced ROP6 nanoclustering also requires phosphatidylserine lipid (negatively charged) and positively charged residues located at the C-terminus of ROP6 in root cells[16]. Because phosphatidylserine synthesis defects alter PC shape, it is likely phosphatidylserine-dependent ROP6 nanocluster provides another layer of modulation to promote ROP6 nanoclustering. In mammalian cells, Ras nanoclustering is regulated by the ratio of cholesterols and phosphatidylserine on the PM[69]. In addition, phosphatidylserine coalesces with cholesterols to support phase separation in the model membrane[70] and is essential for the generation and stabilization of cholesterol-dependent nanoclusters in the live-cell membrane[71]. It is also worth noting that, different Ras isoforms segregate into non-overlapping isoform-specific nanodomains containing distinct lipid composition[69]. Two functionally distinct ROPs, ROP2 and ROP6, though both are activated by auxin, are differently localized to the lobing and indenting regions of the PM to promote lobe and indentation formation, respectively[21,26]. Spatial separation of ROP2 and ROP6 into distinct nanodomains may ensure robust signaling and functional specificity for each ROP. Thus, it will be interesting to investigate whether ROP2 and ROP6 are indeed segregated into two distinct nanodomains with differential dynamic behaviors, and to understand how these potentially distinct nanoclustering behaviors contribute to their specific functions.

Recently, Platre et al. showed that auxin induces phosphatidylserine-mediated ROP6 nanoclustering in Arabidopsis roots[16]. Our findings here and those reported by Platre et al.[16] together show that auxin-induced nanoclustering of cell surface-signaling components plays an important role in the regulation of auxin responses. This PM-localized auxin signaling system enables rapid regulation of cytoplasmic activities and complements the TIR1/AFB-based signaling that mainly controls nuclear gene expression[72,73], thus, greatly expanding the toolbox for auxin signaling machinery that is needed for the wide range of auxin responses. Our results show that auxin-induced ROP6 acylation and TMK1-modulated lipid ordering are critical for the nanoclustering of the cell surface auxin-signaling components. These findings together have firmly established a new auxin signaling mechanism at the cell surface that depends on a localized lipid–protein interaction platform as signaling "hot spot". Interestingly, auxin also promotes the cleavage of intracellular kinase domain from the cell surface TMKs to regulate nuclear gene expression[22]. Therefore, the PM-originated signaling mediated by TMKs is emerging as an important auxin signaling mechanism, and thus it would be interesting to determine whether all TMK-dependent auxin signaling pathways are mediated via nanoclustering.

## Methods

**Growth conditions and plant materials**. *A. thaliana* ecotypes *Columbia-0* (*Col-0*) was used as the wild type in this study. The transgenic plants containing 35S::YFP-*CArop6*, 35S::YFP-*DNrop6*[26], pTMK1::TMK1-GFP[25], or pFER::FER-GFP[74] were reported previously. The *hyd1-E508* seeds were kindly given by Dr. Kathrin Schrick (Kansas State University, USA). The *fk-J79* seeds were gifts from Dr. Jyan-Chyun Jang (The Ohio State University, USA). The *cpi1-1* seeds were kindly shared by Dr. Markus Grebe (Swedish University of Agricultural Sciences, Sweden). The *wei8-1tar2-1* seeds were kindly shared by Dr. Jaimie Van Norman (University of California, Riverside, USA). The flotillin1-mVenus x *tmk1tmk4*, mEGFP-ROP6 x *tmk1tmk4*, flotillin1-mVenus x *ktn1*, TMK1-GFP x *ktn1*, TMK1-GFP x *mor1-1* mutants were generated by genetic crosses and confirmed by genotyping. Plants were grown in soil or on half-strength Murashige and Skoog agar Petri dishes

supplemented with 1% (w/v) sucrose. Controlled environmental conditions were provided in the growth room at 22 °C with 16 h light/ 8 h dark photoperiod.

**DNA constructs and plant transformation**. All constructs were made using the primers listed in Supplementary Table 2. The pGWB601/ROP6::mEGFP-ROP6 plant expression vector was constructed as follows. The *ROP6* promoter (1081-bp upstream of the translational start of *ROP6*) and genomic sequence were amplified from *Col-0* genomic DNA using the following primer pairs: ROP6proF/ROP6proR and ROP6gF/ROP6gR, respectively. The *mEGFP* coding sequence was amplified from the plasmid kindly shared by Dr. Xuemei Chen (University of California Riverside, USA) with the primers mEGFPF and mEGFPR. Those three PCR fragments, *ROP6* promoter, *mEGFP*, and *ROP6* genomic sequence, were fused together by the overlapping PCR. The fused PCR products were recombined into the pDONOR207 vector using BP recombination (Invitrogen), and the resulting entry vector was then transferred into the Gateway binary vector pGWB601 via the LR reaction. The construct with similar design (ROP6::GFP-ROP6) was able to fully complement the *rop6-1* knockout mutant[26]. Because a previous publication reported that the native promoter-driven GFP-tagged AtFlotillin1 had very weak to no fluorescence[37], we generated the native promoter-driven line with the flotillin1 fused with mVenus, which is brighter than GFP. The pGWB661/35S::TagRFP-CArop6 construct was generated as follows. The *CArop6^{Q64L}* sequence was amplified with the primers CArop6F and CArop6R, and then recombined into the pDONOR207 vector using BP recombination (Invitrogen). The resulting entry vector was then transferred into the Gateway binary vector pGWB661 via the LR reaction. The pCAMBIA1300/flotillin1::flotillin1-mVenus vector was generated as follows. The 2247-bp promoter region was amplified from *Col-0* genomic DNA using the primers flotmF/flotmR and subcloned into the pCAMBIA1300 vector via *HindIII* and *BamHI* sites. The genomic sequence (without stop codon) of flotillin1 was amplified using the primers flotgF/flotgR and the mVenus coding sequence was amplified using the primers mVenusF and mVenusR. Those two PCR fragments were further fused by the overlapping PCR. The fused PCR products were subcloned into the pCAMBIA1300 vector using *BamHI* and *KpnI* sites. For the 35S::flotillin1-mCherry construct, the genomic sequence (without stop codon) of flotillin1 was amplified using the primers FCF/FCfusR and the mCherry coding sequence was amplified using the primers FCfusF/FCR. Those two PCR fragments were further fused by the overlapping PCR, and then subcloned into the pCAMBIA1300 vector containing the 35S promoter using *BamHI* and *KpnI* sites. The pCAMBIA1300/flotillin1::LIT6b-mVenus construct was generation as follows. The genomic sequence (without stop codon) of LIT6b (plasma membrane marker) was amplified using the primers LITF+LITmVenusR and the mVenus coding sequence was amplified using the primers LITmVenusF+mVenusR1. Those two PCR fragments were further fused by the overlapping PCR. The fused PCR product was subcloned into the pCAMBIA1300 vector using the *KpnI* site. The pGWB659/ TMK1::TMK1-TagRFP was generated as follows. The TMK1 sequence including the 2616-bp promoter and genomic coding region without the stop codon was amplified using the primers TMK1F/TMK1R, and then inserted into the pDO-NOR207 vector. The resulting entry vector was then transferred into the Gateway binary vector pGWB659 via the LR reaction. The construct with similar design (TMK1::TMK1-GFP) was previously reported[25]. The pCAMBIA1300/UBQ10:: mScarlet-MAP4 construct was generated as follows. Fragments of mScarlet-I, MAP4, and NOS Terminator were amplified using the primer pairs of ScarletF/ ScarletR, MAP4F/MAP4R and NOSF/NOSR, respectively, and then were assembled into a linearized pCAMBIA1300-UBQ10Pro-MCS plasmid by using NEB-uilder HiFi DNA Assembly Master Mix. All constructs were sequence-verified. Stable transgenic lines were generated by using the standard *Agrobacterium tumefaciens*-mediated floral dip method[75] in *Col-0* background.

**Chemicals**. Stock solutions of indole-3-acetic acid (IAA, Sigma-Aldrich) and oryzalin (Sigma-Aldrich) were prepared in dimethyl sulfoxide (DMSO). The stock solution of methyl-β-cyclodextrin (mβCD, Sigma-Aldrich) was prepared in deionized water.

**Analysis of *Arabidopsis* cotyledon pavement (PC) cell shape**. For visualizing PC shape, adaxial cotyledon epidermis of 2–3-day-old seedlings was stained in propidium iodide (2 mg/ml) solution for 20 min and then imaged using a Leica TCS SP5 confocal microscopy with the following settings: excitation 535 nm and emission 600–630 nm. PC images were consistently taken at the mid-region of cotyledons with the same developmental stage. We applied a blind analysis for quantifying PC phenotypes. The researcher who analyzed the shape had no knowledge of the identity of the samples. The number of lobes were counted manually according to the previous method described by Xu et al.[21]. An outgrowth that has a curvature with the depth of at least 1 μm was considered as a lobe, and tricellular junctions were excluded from the lobe counting. The indentation width was determined by measuring the shortest distance across the cell between two indenting regions using ImageJ as previously described[24,27]. The data shown in Figs. 1j, 2f, and Supplementary Fig. 3 are representative of three independent experiments with the same pattern. Note that the homozygous sterol biosynthesis mutants germinate and grow much slower than their wild types, PC phenotypes of

mutants and their wild types were quantified at 7 days after seed plating (Fig. 1i, j; Supplementary Fig. 3A, B).

For quantification of auxin and mβCD effects (Supplementary Fig. 3C), seeds were germinated on solid half-strength Murashige and Skoog (½ MS) medium containing 50 nM IAA, 10 mM mβCD, 50 nM IAA plus 10 mM mβCD, or 0.1% DMSO (mock treatment). A consistent DMSO concentration (0.1%, v/v) was used in all treatments, including mock and mβCD treatments. After germination, liquid ½ MS containing the corresponding chemicals were added onto the solid medium to soak the germinated seeds for additional 2 days. The liquid medium containing the chemicals was refreshed for every 12 h.

**Polarization measurements between lobes and indentations**. Due to the optical resolution limit of confocal microscopy, when cells were analyzed at the cell mid-plane, complementary lobing and indenting PM regions from adjacent cells cannot be resolved as two separate PM membranes. To differentiate the distribution of ordered lipid domains (di-4-ANEPPDHQ staining), filipin–sterol complexes and flotillin1 proteins between lobing and indenting regions, the optical section just above the cell mid-plane (1–1.5 μm) was imaged at which the PM membranes of two adjacent cells were optically separated. The same visualization method was used in Fu et al.[26] and Li et al.[30]. 3D reconstruction of confocal z stacks was carried out in Imaris 9.1.2 (Bitplane).

**ROP6 activity assays**. ROP6 activity assays were performed as described previously[25]. Briefly, protoplasts were isolated from leaves of 3-week-old wild-type plants. Isolated protoplasts were treated with half-strength liquid MS media containing 100 nM IAA for 10 min or 10 mM mβCD for 30 min. Where mβCD was used along with IAA, a pretreatment of 10 mM mβCD was given for 30 min followed by a 10 min co-treatment with mβCD and 100 nM IAA. After treatments, total proteins were extracted from $10^5$–$10^6$ treated protoplasts using extraction buffer (25 mM HEPES, pH 7.4, 10 mM MgCl$_2$, 10 mM KCl, 5 mM dithiothreitol, 5 mM Na$_3$VO$_4$, 5 mM NaF,1 mM phenylmethylsulfonyl fluoride, 1% Triton X-100, and 1× COMPLETE, Mini EDTA-free Protease Inhibitor Cocktail (Roche)). Twenty micrograms of MBP-RIC1-conjugated agarose beads were added into the protein extracts and incubated at 4 °C for 3 h. The beads were washed four times by washing buffer at 4 °C. GTP-bound ROP proteins associated with the MBP-RIC1 beads were boiled for western blotting with rabbit polyclonal anti-ROP6 antibodies (Abiocode) and anti-rabbit IgG-HRP antibodies (GE Healthcare). Prior to the pull-down assay, a fraction of total proteins was analyzed by immunoblot assay to determine total ROP6. The experiments were repeated three times.

**ROP6 *S*-acylation assays**. ROP6 *S*-acylation assays were performed according to the method described by Hemsley et al.[76]. IAA treatments were performed on 3-week-old transgenic plants expressing 35S::GFP-ROP6 by spraying. Leaf tissues were collected 10 min after spraying and immediately flash frozen in liquid nitrogen. Collected samples were ground in liquid nitrogen to a fine powder and then mixed with 1.5 ml lysis buffer (100 mM Tris pH 7.2, 150 mM NaCl, 25 mM EDTA, 2.5% SDS, 25 mM N-Ethylmaleimide) with 1× COMPLETE, Mini EDTA-free Protease Inhibitor Cocktail (Roche). After infiltration through a 200-μm nylon mesh, samples were centrifuged at 16,000 × *g* for 5 min. The supernatant was collected, and protein concentration was determined using the BCA assay. In total, 2 mg of proteins were incubated in the lysis buffer for 2 h at room temperature with gentle mixing and then precipitated using the methanol/chloroform extraction method[77]. The pellets were resuspended in 1 ml of binding buffer (100 mM Tris pH 7.2, 150 mM NaCl, 25 mM EDTA, 2% SDS, 6 M urea with protease inhibitors), and the solution was further divided into two equal aliquots. One aliquot was combined with an equal volume of 1 M hydroxylamine. As a control, the other aliquot was treated identically but hydroxylamine was replaced with 1 M NaCl. After mixing, 50 μl aliquots were taken and used as a loading control. The remaining samples were incubated with 100 μl of a 50% suspension of prewashed Thiopropyl Sepharose CL-6b beads (GE Healthcare) at room temperature for 1 h. The beads were washed three times with 1 ml of binding buffer. Proteins were eluted by incubating the beads at 37 °C for 30 min in 35 μl 2× SDS sample buffer containing 6 M urea and DTT. Samples were analyzed by SDS/PAGE and western blotting using rabbit polyclonal anti-GFP antibodies (Santa Cruz Biotech, Inc) and anti-rabbit IgG-HRP antibodies (GE Healthcare). The experiments were repeated three times.

**Di-4-ANEPPDHQ probe-based membrane ordering measurements**. The di-4-ANEPPDHQ staining was performed according to the procedure previously described[28,29]. The 2–3-day-old seedlings were incubated in 5 μM di-4-ANEPPDHQ dissolved in DMSO for 1 h, washed three times in water, then imaged with a Leica SP5 confocal laser-scanning microscopy (CLSM). The live samples were excited with a 488-nm laser, and fluorescence was detected in two emission windows at 500–580 nm and 620–750 nm. Fluorescence signals were collected by a ×40 oil-immersion objective. Before the dye was added to seedlings, no auto-fluorescence was detected in these two spectral regions when excited with the 488-nm laser light. Plasmolysis with 0.8 M mannitol revealed that the peripheral staining is associated with the plasma membrane, but not with the cell wall (Supplementary Fig. 1). After imaging, we followed the published protocol[29] to

generate pseudo-colored ratiometric general polarization (GP) images. Briefly, the ImageJ macro for GP analysis was applied with the following settings: the threshold value was fixed at 15, the color scale for the output GP images was set to "grays", and no immunofluorescence mask was selected. The GP values were calculated according to the following equations:

$$\text{GP} = (I_{500-580} - GI_{620-750})/(I_{500-580} + GI_{620-750}), \tag{1}$$

$$G = (\text{GP}_{ref} + \text{GP}_{ref}\text{GP}_{mes} - \text{GP}_{mes} - 1)/(\text{GP}_{mes} + \text{GP}_{ref}\text{GP}_{mes} - \text{GP}_{ref} - 1), \tag{2}$$

where $I$ represents the intensity in each pixel in the images collected from two spectral channels of CLSM, 500–580 nm and 620–750 nm; $G$ is a calibration factor, which is used to compensate for different collection efficiency between two channels and is calculated according to Eq. (2); $\text{GP}_{mes}$ is the GP value of di-4-ANEPPDHQ in pure DMSO solution with the same microscopy settings as those used for the real sample. It is calculated using Eq. (1) with $G = 1$; $\text{GP}_{ref}$ is a reference value for di-4-ANEPPDHQ in DMSO, here fixed at −0.85 as suggested[29]. Based on the generated GP images, we further extracted mean intensity values at the complementary lobing and indenting sides of plasma membranes by employing the polygon selection and measurement tools in ImageJ. The extracted values were further normalized with the following equation: GP = (extracted values/127.5) − 1.

For studying the auxin effect on membrane ordering (Fig. 2a–d; Supplementary Fig. 4), seedlings were first treated with 100 nM IAA or 0.1% DMSO (mock treatment) for 1 h, and then concomitantly incubated in 5 μM di-4-ANEPPDHQ for 1 h before imaging.

**Detection of filipin–sterol fluorescence in pavement cells**. The filipin staining was performed according to the procedure previously described[31]. The 2–3-day-old seedlings (wild type or expressing mVenus-tagged proteins) were submerged to 300 μl of microtubule-stabilizing buffer (50 mM PIPES, 5 mM EGTA, 5 mM MgSO4 [pH 7.0]) containing 4% paraformaldehyde (PFA) and 225 μM filipin III (Sigma-Aldrich). Samples were then placed into a microwave oven and pulsed six times for 30 s at 90 W with a 1-min pause in between each pulse. Sample staining/fixation was continued for 1 h at room temperature in the dark. Samples were then washed three times in 1 ml of dH2O and mounted in Citifluor AF1 antifade reagent for observation by a Zeiss LSM880 confocal microscopy. Filipin and mVenus were excited using 364-nm and 488-nm laser lines, respectively. The emission windows were set at 400–485 nm for filipin and 520–560 nm for mVenus. Sequential acquisition was used to avoid fluorescence bleed-through during co-detection of filipin–sterol complexes and flotillin1-mVenus/LIT6b-mVenus.

**Variable-angle total internal reflection fluorescence imaging**. Cotyledons of 2–3-day-old seedlings were imaged on a Zeiss Elyra PS1 system (Zeiss, Germany) with a ×100 Apo (numerical aperture 1.46) oil-immersion objective, in total internal reflection fluorescence (TIRF) mode. The optimum critical angles ranging from 66.8° to 67.5° were used to provide the best signal-to-noise ratio during imaging. Pixel size was 0.107 μm. The development of Arabidopsis cotyledon pavement cells is separated into three stages[24]. To minimize developmental differences, stage II cells with shallow lobes were selected for imaging and the focus was set to the mid-region of pavement cells. Movies were recorded at a rate of ten frames per second (100 ms exposure time) on a 256 × 256-pixel region of interest for 200 frames. Over this imaging period photobleaching was negligible. GFP/mVenus was excited using a 488 nm solid-state laser diode, and the emission was collected with a 495–590-nm band-pass filter and an EM-CCD camera.

For chemical treatments, 2–3-day-old seedlings were incubated in liquid ½ MS medium supplemented with IAA, mβCD, or oryzalin for the indicated time, then imaged within a 10-min time frame window after the treatments. For IAA treatment alone, seedlings were incubated in 100 nM IAA for 10 min, and the mock seedlings were incubated in the medium containing 0.1% DMSO. For the combination of IAA and either oryzalin or mβCD treatments, seedlings were pretreated with 5 μM oryzalin or 10 mM mβCD for 30 min, and then concomitantly treated with 100 nM IAA for 10 min. The mock seedlings were treated with 0.1% DMSO for the same time as the actual treatment. A consistent DMSO concentration (0.1%, v/v) was used in all treatments, including mock and mβCD treatments.

The results shown in Figs. 3–6 and Supplementary Figs. 5 and 9 are representative of two or three independent experiments with the same pattern. In each experiment, at least 35 cells were imaged, representing the minimum of 32,663 particles.

**Colocalization analyses**. For studying the colocalization of flotillin1-mVenus and TMK1-tagRFP, a Zeiss LSM880 equipped with an Airyscan detector was used. Airyscan imaging was performed using 488 nm and 561 nm excitation for mVenus and tagRFP, respectively. Fluorescence emission was collected with a dual 495–550-nm band-pass and 570-nm long-pass filter in combination with a 630 nm short-pass filter. A ×100 Apo (numerical aperture 1.46) oil-immersion objective was used for all imaging. Spearman's rank correlation coefficients were calculated using the Coloc2 plug-in from Fiji software. The negative control was provided by calculating the Spearman's rank correlation coefficient for the same image but after rotating

the image of the red channel with respect to the image from the green channel by 90°, a condition in which only random colocalization should be identified[78]. Same microscope settings were used to image the colocalization of mEGFP-ROP6 and flotillin1-tagRFP, YFP-CArop6 and mScarlet-MAP4, flotillin1-mVenus and mScarlet-MAP4.

**Spot size measurement and single-particle tracking**. Particle-tracking and trajectory analysis were performed using a custom MATLAB R2018a extension code with the Spots Detection module in Imaris 9.1.2 (Bitplane) software. Prior to spot estimation, background subtraction was performed on all images. Estimated spots were modeled with a minimal XY diameter spot size set to 0.3 μm, followed by further filtering using the default "quality" setting automatically calculated by Imaris. Filtered spots were served as seed points for calculating spot sizes using a region growing method. The spot region growth was halted when the local contrast around the border of the spot region met the automatic threshold calculated by Imaris. Spot diameters were then recalculated based on the region volume detected in the previous step. No manual editing was performed.

An autoregressive motion algorithm was used for particle tracking. A maximum distance was set to 0.5 μm. This setting disallows connections between a spot and a candidate match if the distance between the predicted future position of the spot and the candidate position exceeds 0.5 μm. A gap-closing algorithm was used to compensate undetected spots in one or more of the consecutive frames along a spot trajectory to prevent inaccurately fragmented tracks. A maximum gap size, i.e., the allowed maximum number of consecutive time frames with missing spots, was set to eight. Single-particle-tracking trajectories in Supplementary Movies 2, 3, 5, and 6 are shown as dragontail visualization with ten time points of the track length.

**Trajectory analyses**. The mean-square displacement (MSD) and diffusion coefficients were analyzed following a method developed earlier[79]. Trajectories with length >8 frames were selected for MSD and diffusion coefficient analyses. For each track, the MSD ($t$) was calculated from the following formula:

$$\text{MSD}(t) = \frac{1}{L-n} \sum_{s=0}^{l-n-1} (r(s+n) - r(s))^2,$$

where $n = t/\Delta t$, $L$ is the length of the trajectory (number of frames), and $r(s)$ is the two-dimensional position of the particle in frame $s$ ($s = 0$ corresponds to the start of the trajectory). The diffusion coefficient for a spot was determined by fitting a line to MSD ($n\Delta t$) with $n$ ranging from 1 to the largest integer $\leq L/4$[79].

The natural logarithm of diffusion coefficients for all tracks (>8 frames) in each condition were pooled to obtain the log(D) density histogram with the number of bins of 800. The data were then fitted into one-component and two-component Gaussian mixture (univariate, unequal variance) models by using the R package mclust with default settings. The package provides the Bayesian Information Criterion (BIC) to assess how well the model explains the data. For all density distributions of TMK1, flotillin1 and ROP6 particles that were obtained and analyzed in this study, the two-component model yielded significantly higher BIC values than the one-component model, indicating the better fit. Each fit to the two-component model outputs a set of parameters: the mean ($\mu_1$, $\mu_2$), standard deviation ($\sigma_1$, $\sigma_2$) and proportional weights ($\alpha_1$, $\alpha_2$) for each component in the model. The mean ($m_1$, $m_2$) and variance ($v_1$, $v_2$) of the lognormal distribution for each component is calculated based on the following equations:

$$m = \exp(\mu + \sigma^2/2)$$

$$v = \exp(2\mu + \sigma^2)(\exp(\sigma^2) - 1).$$

The overall average of diffusion coefficient ($m_{overall}$) and variance ($v_{overall}$) is calculated based on the following equations:

$$m_{overall} = m_1*\alpha_1 + m_2*\alpha_2$$

$$v_{overall} = \alpha_1*v_1 + \alpha_1*(m_1 - m_{overall})^2 + \alpha_2*v_2 + \alpha_2*(m_2 - m_{overall})^2.$$

The calculated $m_1$, $m_2$, and $m_{overall}$ correspond to $D_0$, $D_1$, and $D_{overall}$ in Supplementary Table 1. Violin plots in the main and Supplementary Figs. (Figs. 3f, 4f, 5c, 5f, 5h, 6c, 6f; Supplementary Fig. 9B, D, F, I) show the distribution of the original diffusion coefficients without the log transformation. Distributions of log-transformed datasets are shown in Supplementary Fig. 5. The overall diffusion coefficients were used to determine statistical significance of treatment effects.

**Quantification and statistical analyses**. All data analyses were performed with PRISM software (GraphPad Software Inc., La Jolla, CA). Each sample was subjected to three different normality tests (D'Agostino & Pearson, Shapiro–Wilk, and Kolmogorov–Smirnov) to determine whether the sample is normally distributed. Samples were considered as a normal distribution as long as they passed one of the three normality tests ($P = 0.05$). Accordingly, parametric ($P \geq 0.05$) or nonparametric tests ($P < 0.05$) were performed. For parametric test, the paired $t$ test or the unpaired $t$ test with Welch's correction was used to assess statistical difference between paired or unpaired samples when datasets contain only two samples, while analysis of variance (ANOVA) coupled with multiple comparison was used for datasets containing more than two samples. For nonparametric test, Wilcoxon

matched-pairs signed-rank test was used for datasets containing paired samples and the Kruskal–Wallis test followed by Dunn's multiple comparisons test was used for datasets containing more than two samples. $*P < 0.05$, $**P \leq 0.01$, $***P \leq 0.001$, $****P \leq 0.0001$; ns, not significant.

The following tests were performed for each graph in the main and Supplementary Figs.:

(1) Wilcoxon matched-pairs signed-rank test, significance level 5%: Fig. 1e, Fig. 1h, and Supplementary Fig. 2B.
(2) Paired $t$ test, significance level 5%: Fig. 3h.
(3) Unpaired $t$ test with Welch's correction, significance level 5%: Fig. 1j, Fig. 5c, and Supplementary Figs. 3B, 6C, 9A–D.
(4) One-way ANOVA with Tukey's test, significance level 5%: Fig. 2d.
(5) Two-way ANOVA with Tukey's test, significance level 5%: Fig. 3c, Fig. 3f, Fig. 4c, Fig. 4f, Fig. 5f, Fig. 5h, Fig. 6c, Fig. 6e, Fig. 6f, Supplementary Fig. 9F, I.
(6) Two-way ANOVA with Fisher's LSD test, significance level 5%: Supplementary Fig. 4B, 9E, H.
(7) Kruskal–Wallis test with Dunn's test, significance level 5%: Fig. 2b, Fig. 2f, Fig. 2h, Fig. 3e, Fig. 4d, Fig. 5e, Fig. 5g, Fig. 6b, and Supplementary Fig. 3C.

The detailed statistical results of each figure are shown in the Source Data file.

**Reporting summary**. Further information on research design is available in the Nature Research Reporting Summary linked to this article.

## Data availability
All data underlying this study are available from the corresponding author upon request. Source data are provided with this paper.

## Code availability
All code that support the findings of this study are available from the corresponding author upon request.

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

## Acknowledgements

We are grateful to Dr. Yvon Jaillais (Univ Lyon) for his critical reading of this paper and helpful suggestions on this work. We thank Dr. Carolyn Rasmussen (UCR) for valuable discussions on this work. We thank members of the Yang laboratory for their technical assistance and helpful discussions. We appreciate access to and microscopy assistance from the Institute of Integrative Genome Biology Microscopy Core Faculty (UC Riverside) and David Carter. X.P. was a Natural Sciences and Engineering Research Council of Canada (NSERC) post-doctoral fellowship holder.

## Author contributions

The project was conceived and supervised by Z.Y. X.P. conceived the study, designed and performed all experiments, except for ROP activity assays. L.F. and U.M. provided the MATLAB code for single-particle tracking. J.L. developed the Python code for batch data processing and analyzed particle-tracking data. J.L., B.S.A., and W.C. developed the MATLAB code for diffusion-coefficient analyses. W.L. conducted ROP activation assays. S.Z. assisted in the genotyping experiments and pavement cell phenotype quantification. T.Z. assisted in the TIRF imaging. J.G. generated the *mScarlet-MAP4* line. Z.Y. and X.P. wrote the paper. J.V.N. helped with the paper revision and provided valuable suggestions.

## Competing interests

The authors declare no competing interests.
