## [Peer Review File · Nature Communications]

Reviewers' comments:

Reviewer #1 (Remarks to the Author):

The authors address the mechanisms underlying cell morphogenesis of the leaf epidermis and investigate differential organization of the plasma membrane in lobes and indents of pavement cells. The study demonstrates an auxin-induced and sterol-dependent clustering of the receptor-like kinase TMK1 and of the small GTPase ROP6 at the membrane and establishes a model that integrates diffusional regulation of ROP6 through membrane nanodomains and cortical microtubules to explain symmetry breaking and cell polarization in this cell type. The study shows some parallels to a recently published work by Platre et al. 2019, who demonstrated auxin- and phosphatidylserine-dependent recruitment of active ROP6 into membrane nanodomains in root epidermis cells during regulation of cell expansion. The present manuscript by Pan et al. confirms some findings made by Platre et al. (such as the auxin-driven clustering of ROP6), but also contains a number of aspects that clearly distinguish the two papers.

In general, the presented data is convincing in most cases (see specific points below). I have, however, major concerns regarding some interpretations and conclusions.

A main critical point is that the authors aim to understand cell polarization, but never actually investigate a polarizing cell. Besides the theoretical cells in their computational models, all here analyzed cells are well beyond the point of polarization. For example, the fact that lobes and indents show different degrees of membrane order, differential distribution of certain proteins or lipids, does not automatically mean that these asymmetries drove lobe/indent formation in the first place. To make solid statements about causalities, cells should be investigated before lobe formation, as the same group has shown it in a previous study (Fu et al. 2005). It is well possible that what the authors observe is a consequence, not cause of PC morphogenesis.

In fact, indentation is never abolished in any of the conditions. The number of lobes and the extent of lobe elongation is reduced. The fact that they are still present speaks, however, for regulation of directional growth downstream of the actual polarization.

Specific points:

Pg 3, l 31. The authors should be careful to not adopt an outdated narrative of "lipid rafts" (see Munro, Cell, 2003; Lichtenberg et al. Trends Biochem Sci, 2005; Malinsky et al., Annu Rev Plant Biol, 2013, and others).

The authors use flotillin-1 as "well-established marker for ordered sterol-rich nanodomains". This is not the case. In the early days of "lipid rafts", flotillins have been identified in detergent-resistant membranes and despite efforts in several early works to equate DRMs with biomembranes in vivo, flotillins have, to my knowledge, never been unequivocally shown to occupy ordered domains or associate with sterols (except after Triton-X treatment).

Along the same lines, from the fact that sterols are important for PC morphogenesis, one cannot jump to the conclusion that the involved proteins are to be found in "sterol-rich ordered lipid domains". Sterols make up roughly 20% of the plasma membrane lipid composition - it is not surprising that protein dynamics and protein functions are perturbed under depleting conditions, be it genetically or pharmacologically (m β CD). Studies using NanoSIMS have also debunked the notion that sterols are always associated with liquid-ordered membranes (e.g. see Frisz et al JBC 2013). Sterols are important structural elements of membranes and precursors to hormones, hence, sterol biosynthesis mutants typically exhibit major pleiotropic growth defects that are not mentioned in the present study but likely affect cell shapes well beyond PCs. The authors need to take these wider implications into account to not be accused of pursuing a cherry-picking approach.

Fig 1. For their ANEPP staining, the authors show images of cells that are clearly not imaged and analyzed at the cell mid-plane, which is not mentioned anywhere. How can this be explained or justified? In cells where mid-plane sections are shown (Fig 2), no GP polarity can be observed.

This raises serious doubts and should be addressed.

Instead of using flotillins as marker for sterols (which is baseless, see above), the authors could try filipin staining of sterols.

TIRF images: again the authors do not mention a critical detail: the fact that all TIRF data are most likely taken on periclinal membranes, while the analyses above concerned anticlinal cell boundaries. From the figures it is not possible to tell where in the organ or the cell these images were taken. Why are the authors not mentioning that these are different parts of the cell? This is highly relevant as the authors suggest that different environments exist within the plasma membrane.

Also, is it really correct that TIRF was used here, and not Variable Angle Epifluorescence Microscopy (Konopka & Bednarek, Plant J 2008)? Normal TIRF should not be possible on the leaf surface (cell wall too thick). The method part also indicates that illumination angles were individually chosen to optimize s/n ratio. This, of course, opens up possibilities for experimental bias and should be well controlled.

Could differences in protein motilities in lobes and indents be show through FRAP experiments?

Flot1-mCherry and Flot1-mVenus largely differ in their localization patterns (Fig 3, 4). How can this be explained? Why is it not discussed?

Fig 3G. The colocalization analysis is not convincing as this partial colocalization could be coincidence. Please provide a statistical analysis of multiple biological replicates and compare correlation coefficients with control measurements after rotating one channel.

Fig 5A. I find it surprising that DNrop6 shows nanoclusters at the PM. Studies from the lab of Shaul Yalovsky (e.g. Lavy et al (Curr Biol 2007)) have shown that dominant negative ROPs are mostly cytosolic. Can you please explain what you see there? Have you imaged a non-transgenic wild type plant to test whether some of this signal could be attributed to autofluorescence?

Another issue with the CA- and DN-ROP6 data is the use of overexpression using 35S. This issue is again not discussed in the main text. How can these data be compared? Could line-specific differences in expression levels influence the results?

Please use dot-plots or violin-plots instead of bar graphs (Fig 1F, 2E, 3F, 4F, 5B,E,G, 7C,F).

In general, I am convinced that several findings of this study have great value and should be published. However, the authors should consider to rethink their conclusions, refrain from overstatements and openly discuss any shortcomings or limitations e.g. of the ANEPP staining, the use of mBCD or the limited accessibility of anticlinal PM for imaging. In its current state, the manuscript leaves the impression as if the authors try to push for a certain hypothesis and hide some potential points of criticism. With more careful interpretations and critical discussion this study will become a valuable contribution to our understanding of membrane organization and cell polarization.

Minor points:

Introduction: Instead of citing in bulk, the authors should elaborate better the state of the art regarding mechanisms of membrane organization and models for symmetry breaking. Some explanations (e.g. of "lipid rafts") are too simplistic or outdated and require a more differentiated introduction. Since the introduction is rather superficial, the specific research questions that drive this study are also not well worked out.

Pg 10, line 42. This sentence should probably be moved to the beginning of the paragraph.

Reviewer #2 (Remarks to the Author):

In this manuscript, the authors claimed that nanoclustering of cell surface transmembrane receptor kinase 1 drives the jigsaw-puzzle pattern of plant leaf epidermal cells. The authors focused on the ordered lipid in the formation of polarity in a single cell and examined the functions of the related molecules using live imaging techniques. They also formulated a mathematical model of the polarity formation and provided some experimental evidence to support their claim. While the experimental part seems to be a substantial contribution to this field, mathematical modeling does not represent the actual situation (although carefully written) and does not add significant insight into this work. This problem comes from miscommunication between experimental biologists and mathematicians in the group. The authors should rewrite the mathematical model from scratch or completely remove the mathematical modeling part.

* Their model only deals with the dynamics of molecules on the single plasma membrane, which does not represent the actual experimental situation. In the plant leaf, two plasma membranes sandwich the cell wall of constant thickness, and the indentation region in a cell is the protrusion region in the neighboring cell. The model ignored this aspect. The model should incorporate this aspect, or the experimental biologists in the authors should show experimental results in which isolated single epidermal cells can generate indentation under some condition.

* There are at least three previous works that deal with spontaneous pattern formation of plant leaf epidermal cells (Higaki et al., 2017, 2016; Sapala et al., 2018). Two of them utilize buckling instability of the cell wall, and one utilizes interface instability of band-like solution in a reaction-diffusion system, which implements production and degradation of cell wall controlled by diffusible molecules. The references to these works are missing. The authors should discuss the relationship between these previous models and their model.

* The patterns their model generated are not similar to the actual pattern (Fig. 6D, E). The actual patterns consist of smooth curves and seem to have some characteristic length, while their simulation results seem to be random clefts. I want to confirm whether the experimental biologists in the author group are satisfied with this result.

* Mathematical analysis and insight of the dynamics of their "mathematical" model are missing. There exists a standard mathematical analysis of why pattern formation takes place (linear stability analysis), and that will help to understand what wavenumber should be selected (=size and number of the structures). Their model is a slightly modified reaction-diffusion model, and it should be easy to reduce the model to an analytically manageable form. Mathematicians in the authors should do more mathematics, not dumbing down the ideas to communicate with biologists. Currently, the model only numerically shows that with a specific parameter set, the model behavior looks similar. At first, they should formulate a new model which includes the plasma membranes of two adjacent cells and cell walls, and reduce the model to undertake mathematical analysis to understand the model behavior, and possibly provide a reliable prediction on the manipulation of the pattern formation. Otherwise, just removing the modeling part may be a good option because the experimental part is good enough to stand alone.

Higaki, T., Kutsuna, N., Akita, K., Takigawa-Imamura, H., Yoshimura, K., Miura, T., 2016. A Theoretical Model of Jigsaw-Puzzle Pattern Formation by Plant Leaf Epidermal Cells. *PLoS Comput. Biol.* 12, e1004833. <https://doi.org/10.1371/journal.pcbi.1004833>

Higaki, T., Takigawa-Imamura, H., Akita, K., Kutsuna, N., Kobayashi, R., Hasezawa, S., Miura, T., 2017. Exogenous cellulase switches cell interdigitation to cell elongation in an RIC1-dependent manner in *Arabidopsis thaliana* cotyledon pavement cells. *Plant Cell Physiol.* 58, 106–119.

<https://doi.org/10.1093/pcp/pcw183>

Sapala, A., Runions, A., Routier-Kierzkowska, A.L., Gupta, M. Das, Hong, L., Hofhuis, H., Verger, S., Mosca, G., Li, C.B., Hay, A., Hamant, O., Roeder, A.H.K., Tsiantis, M., Prusinkiewicz, P., Smith, R.S., 2018. Why plants make puzzle cells, and how their shape emerges. *Elife* 7, 1–53.

<https://doi.org/10.7554/eLife.32794>

Reviewer #3 (Remarks to the Author):

This manuscript by Pan et al describes how nano-clustering of membrane proteins contributes to cell polarization. Arabidopsis pavement cells form multiple indentations during cell growth. Authors' group previously revealed that auxin signaling promotes formation of the indentations through the TMK1 receptor kinase, ROP6 GTPase, and microtubules. In the present study, authors show that TMK1 and ordered sterol lipids are clustered upon auxin signaling, which in turn promotes ROP6 activation and their clustering. Furthermore, by using mathematical models, authors predicted that additional feedbacks from microtubules, inhibiting diffusion of TMK1 and of ordered sterol lipids are required for the nano-clustering-induced cell polarization. Finally, authors show that microtubules indeed influence the clustering of TMK1 and ordered lipid sterols. Generally, the research is well organized, data are of high quality, and discussion is thoughtful. This elegant study provides novel insights into the role of nano-clustering for cell polarization, significantly updating our understanding of how cell polarization is achieved through regulating the dynamics of plasma membrane proteins. I have several suggestions.

1. Previously, authors' group revealed that the ROP2/4-actin pathway promotes formation of lobes, acting as a counterpart of the ROP6 pathway. It would be very interesting if authors discuss specificity of nanoclustering for ROP6 and others ROPs. This may provide readers with useful information to grasp the overview of pavement cell morphogenesis.

2. Authors observed mEGFP-ROP6 and TMK1-GFP expressed in wild type plants, so it is unclear whether these fusion proteins are functional. I suggest authors mention that these (or similar) constructs complemented mutants in previous studies.

3. I was a bit confused because authors added a feedback from ROP6 to TMK1 and lipids after showing co-localization of CA-ROP6 and microtubules (Fig S6). I suppose that authors conceived microtubules trap TMK1 and lipids, which in turn promotes clustering of active ROP6, resulting in the accumulation of active ROP6 along microtubules. It would be better to add some explanation to connect the result and hypothesis.

In this case, however, I think it is also possible that the microtubules restrict the diffusion of active ROP6 directly or indirectly via RIC1 or other microtubule-associated proteins, then active ROP6 does promote clustering of TMK1 and sterols along microtubules. Could authors discuss or examine this point? To clarify the relationship between ROP6, microtubules, and TMK1, it may be useful if authors examine whether CArop6 expression affects TMK1/flotillin1 dynamics in the presence of oryzalin. Also, I suggest that authors observe co-localization of ROP6 and TMK1/ flotillin1 to confirm their association in the lipid/protein clusters.

4. To confirm the feedback from microtubules to TMK1 and flotillin1, authors analyzed TMK1 and flotillin1 dynamics upon treatment with oryzalin. Although this nicely demonstrated that microtubules indeed influence the clustering of TMK1 and flotillin, the experimental condition seems not to reflect the exact downstream events of ROP6 where ROP6 promotes microtubule ordering through RIC1 and katanin activities. It would be interesting if authors examine dynamics of TMK1 and flotillin1 in *ric1* or katanin mutants or overexpression lines. I think this makes the story more robust. Alternatively, authors may test if taxol promotes microtubule ordering and influences dynamics of TMK1 and flotillin1 in young pavement cells.

5. The *tmk1tmk4* mutant exhibits severer developmental phenotype. It would be possible that the phenotypes of *ROP6* and *flotillin1* in *tmk1tmk4* mutant includes secondary effects of the developmental defects. To eliminate this possibility, I suggest authors test transient treatment with auxin transport inhibitor NPA.

Minor

Page 6, line 34 "Jaillais's group showed that~"

Here, another phrase may be preferred such as "Platre et al. (2019) showed that" or "it was shown that" or others.

Page 8, line 26

It is better to remove "a high concentration of".

Page 8, line 26

Remove a space from "flotillin 1".

Page 51, GPF should be GFP.

Responses to referees

Reviewers' comments:

Reviewer #1 (Remarks to the Author):

The authors address the mechanisms underlying cell morphogenesis of the leaf epidermis and investigate differential organization of the plasma membrane in lobes and indents of pavement cells. The study demonstrates an auxin-induced and sterol-dependent clustering of the receptor-like kinase TMK1 and of the small GTPase ROP6 at the membrane and establishes a model that integrates diffusional regulation of ROP6 through membrane nanodomains and cortical microtubules to explain symmetry breaking and cell polarization in this cell type. The study shows some parallels to a recently published work by Platre et al. 2019, who demonstrated auxin- and phosphatidylserine-dependent recruitment of active ROP6 into membrane nanodomains in root epidermis cells during regulation of cell expansion. The present manuscript by Pan et al. confirms some findings made by Platre et al. (such as the auxin-driven clustering of ROP6), but also contains a number of aspects that clearly distinguish the two papers.

In general, the presented data is convincing in most cases (see specific points below). I have, however, major concerns regarding some interpretations and conclusions.

We thank the reviewer for the positive and constructive comments!

A main critical point is that the authors aim to understand cell polarization, but never actually investigate a polarizing cell. Besides the theoretical cells in their computational models, all here analyzed cells are well beyond the point of polarization. For example, the fact that lobes and indents show different degrees of membrane order, differential distribution of certain proteins or lipids, does not automatically mean that these asymmetries drove lobe/indent formation in the first place. To make solid statements about causalities, cells should be investigated before lobe formation, as the same group has shown it in a previous study (Fu et al. 2005). It is well possible that what the authors observe is a consequence, not cause of PC morphogenesis. In fact, indentation is never abolished in any of the conditions. The number of lobes and the extent of lobe elongation is reduced. The fact that they are still present speaks, however, for regulation of directional growth downstream of the actual polarization.

Answer: We thank the reviewer for pointing out this concern. As suggested, we analyzed the sterol distribution and the auxin effects in the stage *I* cells in cotyledons (Supplementary Figure 8), which have no obvious lobes and indentations (1 day after germination). As shown in Supplementary Figure 8A, the patchy distribution of sterol-rich domains was also observed in Stage *I* cells, suggesting that lateral segregation (or polar distribution) of sterol-rich domains is not a consequence of PC morphogenesis. In addition, the auxin-induced increase in size and decrease in diffusion rate of flotillin1 and TMK1 nanoclusters were also shown in Stage *I* cells (Supplementary Figures 8B-8C). These results strongly support our hypothesis that TMK1/sterol nanoclustering and auxin triggered dynamic change is the cause but not the result of pavement cell polarity. To further test this hypothesis, it would be nice to directly image the entire initiation event from auxin-induced nanoclustering of TMK1/sterols to the formation of laterally

segregated polar domains. However, this experiment is technically infeasible. Therefore, we used the mathematical model to simulate the process. As our model shows auxin triggered nanoclustering in combination with the MT-dependent feedback mechanism is sufficient for the establishment of the lateral segregation of signaling complexes at the plasma membrane (Figure 6). This model was further validated by our experimental data (Figure 7 and Supplementary Figure 11). Finally, we have shown that the sterol biosynthesis mutant which is insensitive to auxin-induced lipid ordering at the PM compromised the promotion of lobe or indentation numbers by auxin (Figures 2C-2E). Taken together our results provide strong evidence that auxin promotes the establishment of cell polarity directly through its effect on TMKs/ROP6 nanoclustering.

We believe that the incomplete loss of indentations in the mutants studied here does not argue against the importance of nanoclustering of TMKs and ROP6 in cell polarity establishment but was due to functional redundancy of the relevant genes. None of the mutants used in our study were null, e.g., the *tmk1/4* double mutant contains TMK2 and 3 that are functionally overlapping with TMK1/4, while null mutants (e.g., *tmk1234*) are lethal and not usable for our analyses. Similarly, the sterol perturbation either via pharmaceutical or genetic approaches does not lead to the complete removal of sterols.

Specific points:

Pg 3, l 31. The authors should be careful to not adopt an outdated narrative of "lipid rafts" (see Munro, Cell, 2003; Lichtenberg et al. Trends Biochem Sci, 2005; Malinsky et al., Annu Rev Plant Biol, 2013, and others). The authors use flotillin-1 as "well-established marker for ordered sterol-rich nanodomains". This is not the case. In the early days of "lipid rafts", flotillins have been identified in detergent-resistant membranes and despite efforts in several early works to equate DRMs with biomembranes in vivo, flotillins have, to my knowledge, never been unequivocally shown to occupy ordered domains or associate with sterols (except after Triton-X treatment). Instead of using flotillins as marker for sterols (which is baseless, see above), the authors could try filipin staining of sterols.

Answer: We thank the reviewer for this valuable comment. As suggested, we have performed filipin-sterol labeling. Following the published protocol¹, we first conducted filipin-sterol labeling of live seedling samples. We observed the polar enrichment of filipin-sterol fluorescence at the emerging tip of the root hair as previously reported² and flotillin1-mVenus expressed from its own promoter showed similar distribution pattern (Fig R1, Below). However, this protocol was unable to stain the cotyledon pavement cells due to inability of filipin to penetrate the cotyledon epidermal cells. Therefore, we next followed the protocol for filipin-sterol labeling of PFA fixed samples¹. As shown in Supplementary Figures 2A-2C and Supplementary Video 1, filipin staining showed a patchy distribution of filipin-sterol complexes at the PM of pavement cells and these patchy sterol complexes had an indentation preferred localization. This is consistent with the ANEPPDHQ staining and flotillin1-mVenus distribution. Because live cell staining does not work for the cotyledons, we were unable to use filipin to study the dynamics of sterol-rich nanodomains up on auxin treatments. Therefore, we performed colocalization analysis of filipin-sterol complexes and flotillin1-mVenus to test whether flotillin1 is a good marker for sterol-rich nanodomains. The results showed that the distribution of

flotillin1-containing nanodomains were well colocalized with the patched filipin-sterol complexes at the plasma membrane (Supplementary Figures 2D-2G). In contrast to flotillin1 and filipin, we did not observe colocalization between the plasma membrane marker LIT6b and filipin-sterol complexes. Consistently, flotillin were previously found to co-localize and associate with cholesterol-rich microdomains not only in biochemical studies, but also in microscopy studies³⁻⁵. Based on these results, we believe that flotillin can be used as a valid marker to study the dynamics of sterol-rich nanodomains.

Fig R1. Co-detection of filipin-sterol complexes and flotillin1-mVenus at the emerging tip of the root hair. Left panel: the bright-field image. Middle panel: flotillin1-mVenus fluorescence. Right panel: filipin-sterol fluorescence.

Along the same lines, from the fact that sterols are important for PC morphogenesis, one cannot jump to the conclusion that the involved proteins are to be found in "sterol-rich ordered lipid domains". Sterols make up roughly 20% of the plasma membrane lipid composition - it is not surprising that protein dynamics and protein functions are perturbed under depleting conditions, be it genetically or pharmacologically (m β CD). Studies using NanoSIMS have also debunked the notion that sterols are always associated with liquid-ordered membranes (e.g. see Frisz et al JBC 2013). Sterols are important structural elements of membranes and precursors to hormones, hence, sterol biosynthesis mutants typically exhibit major pleiotropic growth defects that are not mentioned in the present study but likely affect cell shapes well beyond PCs. The authors need to take these wider implications into account to not be accused of pursuing a cherry-picking approach.

Answer: We thank and agree with the reviewer for the critique that sterol perturbation either by genetic or pharmacological approaches can have pleiotropic effects not necessarily linked to the sterol-rich nanodomains. Thus, we added a sentence stating that the deficiency in sterol accumulation may generate pleiotropic effects but the defect in PC morphogenesis most likely resulted from the alteration of ordered lipid nanodomain, as supported by other observations described in this paper (Figs. 1, 2A-2D, 3G and Supplementary Figure 2). Although non-specific effects could not be excluded, all of our results support the specific involvement of sterols in the auxin-induced formation of ordered lipid nanodomains and morphogenesis in PCs, including the similar PC shape changes (i.e., increased indentation widths and reduced lobe numbers) resulting from defects in ROP6 signaling and sterol accumulation, the colocalization of flotillin1

nanodomains with the filipin-sterol complexes, partial colocalization of TMK1 and flotillin1, auxin triggered lipid ordering and flotillin dynamic change point. We also included this discussion in the revised manuscript (Page 4, lines 35-42).

Fig 1. For their ANEPP staining, the authors show images of cells that are clearly not imaged and analyzed at the cell mid-plane, which is not mentioned anywhere. How can this be explained or justified? In cells where mid-plane sections are shown (Fig 2), no GP polarity can be observed. This raises serious doubts and should be addressed.

Answer: We thank the reviewer for this helpful comment. Due to the optical resolution limit of confocal microscopy, when cells were analyzed at the cell mid-plane, complementary lobe and indentation regions on adjacent sides of the cell cannot be resolved as two separate complementary plasma membranes. However, the complementary lobe and indentation PM regions can be resolved to observe the non-uniformed distribution of ordered domains (ANEPP staining), filipin-sterol complexes, and flotillin1 in cells, when the optical sections just above (1-1.5 μm) the cell mid-plane was used during imaging. The same visualization method was used in previous studies^{6,7}. A description of this imaging protocol has been added into the text (Page 3, Line 42-45) and method section (Page 15, Lines 1-8). In Figure 2, the cells were visualized in the mid-plane to quantify the effect of auxin on the overall levels of lipid ordering across the PM. For easy quantification, we did not separate lobe and indentation regions for this experiment. This explanation has also been added into the legend of Figure 2. Please see Page 32, Lines 1-3.

TIRF images: again the authors do not mention a critical detail: the fact that all TIRF data are most likely taken on periclinal membranes, while the analyses above concerned anticlinal cell boundaries. From the figures it is not possible to tell where in the organ or the cell these images were taken. Why are the authors not mentioning that these are different parts of the cell? This is highly relevant as the authors suggest that different environments exist within the plasma membrane.

Answer: We agree with the reviewer's point. We added a schematic of the geometry of the cotyledon epidermal cell to explain the region of cell imaged (Fig. 3A). As the reviewer pointed out, the periclinal membrane of the PC was analyzed for protein dynamics. This information is now provided in the text (Page5, lines 43-46). Unfortunately, none of the current methods (including TIRF and PALM) for imaging nanoclusters allow the analysis in the periclinal region. However, we believe that our data is relevant to our hypothesis as we analyzed the effect of auxin on the dynamic behavior of nanoclusters.

Also, is it really correct that TIRF was used here, and not Variable Angle Epifluorescence Microscopy (Konopka & Bednarek, Plant J 2008)? Normal TIRF should not be possible on the leaf surface (cell wall too thick). The method part also indicates that illumination angles were individually chosen to optimize s/n ratio. This, of course, opens up possibilities for experimental bias and should be well controlled.

Answer: We thank the reviewer for the comment. Imaging was performed on a Zeiss Elyra PS1 system. The recent publication used the same TIRF module for the super-resolution PALM

imaging in plant roots⁸. In VAEM, sub-critical angles (59–61°) are used for the incident beam^{9,10}. However, in our experiment, the used angles are ranging from 66.8° to 67.5° within the indicated TIRF mode of the Zeiss Elyra PS1 system. In addition, the used angles are within the suggested angle range (65.2°-72.5°) for true-TIRF imaging in plant tissues⁹. Also, 2-3 days-old cotyledons rather than leaves were used in our imaging. The cotyledons should have thinner cell wall than leaf tissues. Similar to the published study, the optimal critical angle was determined as giving the best signal-to-noise ration⁸. We did not adjust angles for individual samples. Instead, the same angle was consistently used for samples within the same batch. But slightly different angles (66.8° to 67.5°) have to be used for samples from different batches, possibly due to growth variations at the level of cell wall thickness between batches. In the revised manuscript, we have specified the angle range used in our study and updated the term to VA-TIRFM (Page17, Lines 4-8).

Could differences in protein motilities in lobes and indents be show through FRAP experiments?

Answer: We thank the reviewer for this suggestion. We tried the FRAP experiments on flotillin1-mVenus. To exclude the recovery artifacts caused by endo- or exocytotic removal or insertion of protein from the membrane¹¹, we monitored the values of fluorescence recovery at 2 mins. In contrast to LIT6b-mVenus, flotillin1-mVenus displayed extremely slow FRAP kinetics (Fig. R2). This is consistent with the slow diffusion rate of the flotillin1 particles. Because of this slow recovery, we were unable to detect the differences in protein motilities between lobes and indents using the FRAP. Therefore, the temporal resolution (seconds) of FRAP limits its application to study the mobility differences of flotillin1-mVenus between lobes and indents.

Fig. R2. Flotillin-mVenus has low mobility. Left panel: FRAP examples illustrating slow (flotillin1-mVenus) and fast (LTI6b-mVenus) membrane lateral diffusion. Right panel: FRAP curves of mVenus fusion proteins. FRAP experiments were carried out on Arabidopsis pavement cells expressing flotillin1::flotillin1-mVenus or flotillin1::LTI6b-mVenus. Scale bar: 2 μ m.

Flot1-mCherry and Flot1-mVenus largely differ in their localization patterns (Fig 3, 4). How can this be explained? Why is it not discussed? Fig 3G. The colocalization analysis is not convincing as this partial colocalization could be coincidence. Please provide a statistical analysis of multiple biological replicates and compare correlation coefficients with control measurements after rotating one channel.

Answer: We thank reviewer for the critiques and suggestions. We are not sure about the exact cause of the different localization pattern, presumably caused by the microscopic system (ONI Nanoimager) we used previously. In the revised manuscript, we used the new transgenic line co-expressing *flotillin1::flotillin1-mVenus* and *TMK1::TMK1-tagRFP*. We tried the dual color imaging by using the Zeiss Elyra PS1 system but failed to obtain reliable images due to high level of crosstalk between channels. Therefore, we used another available system, Zeiss 880 Airyscan confocal microscopy, which was recently used to study plant plasma-membrane protein nanodomains¹². As shown in Fig. 3G-3H, *TMK1-tagRFP* particles are at least partially colocalized with *flotillin1-mVenus* particles. To test the specific colocalization, the merged image was generated by rotating the image from the red channel with respect to the image from the green channel by 90°, a condition in which only random colocalization can be observed¹³. The significantly higher correlation coefficients obtained in nonrotated images than in 90° rotated images, indicating that the observed colocalization is not due to random chance. The details are described in the legend of Figure 3 (Page 34, Lines 16-22) and in the method section (Page 17, Lines 17-27).

Fig 5A. I find it surprising that DNrop6 shows nanoclusters at the PM. Studies from the lab of Shaul Yalovsky (e.g. Lavy et al (Curr Biol 2007)) have shown that dominant negative ROPs are mostly cytosolic. Can you please explain what you see there? Have you imaged a non-transgenic wild type plant to test whether some of this signal could be attributed to autofluorescence? Another issue with the CA- and DN-ROP6 data is the use of overexpression using 35S. This issue is again not discussed in the main text. How can these data be compared? Could line-specific differences in expression levels influence the results?

Answer: We thank the reviewer for the comments. The GFP-DNrop6^{D121A} line used in our study is different from the line (GFP-DNrop6^{T30N}) used by Yalovsky's group¹⁴. Unlike DNrop6^{T30N}, the recruitment of GFP-DNrop6^{D121A} to the PM is not compromised and do not accumulates in nuclei (Supplementary Figure 6). The signals obtained in TIRF imaging is not the autofluorescence as the non-transgenic wild type plant does not display similar signals. To compare the dynamics of CA- and DN-ROP6, we chosen the CArop6 and DNrop6 lines with weak but similar expression level (Supplementary Figure 6). Based on these, we believe the CA- and DN-rop6 lines are comparable.

Please use dot-plots or violin-plots instead of bar graphs (Fig 1F, 2E, 3F, 4F, 5B, E, G, 7C, F).

Answer: We thank the reviewer for the suggestion. The figures have been revised.

In general, I am convinced that several findings of this study have great value and should be published. However, the authors should consider to rethink their conclusions, refrain from overstatements and openly discuss any shortcomings or limitations e.g. of the ANEPP staining,

the use of m β CD or the limited accessibility of anticlinal PM for imaging. In its current state, the manuscript leaves the impression as if the authors try to push for a certain hypothesis and hide some potential points of criticism. With more careful interpretations and critical discussion this study will become a valuable contribution to our understanding of membrane organization and cell polarization.

Answer: We thank the reviewer for all of the valuable comments. As suggested, we have rephrased some of our conclusions and discussed the shortcomings of certain experimental approaches. Specifically, in the revised manuscript, the shortcomings of the usage of m β CD and sterol-synthesis mutants have been discussed. Please see (Page 4, lines 35-42). In addition, the filipin staining was added to complement the ANEPP staining in our revision (Supplementary Figure 2). Furthermore, a schematic of the geometry of the cotyledon epidermal cell (Fig. 3A) along with the discussion of the limited accessibility of anticlinal PM for imaging was described in the text (Page 5, lines 43-46). We again appreciate the reviewer's comments that helped us better present our work.

Minor points:

Introduction: Instead of citing in bulk, the authors should elaborate better the state of the art regarding mechanisms of membrane organization and models for symmetry breaking. Some explanations (e.g. of "lipid rafts") are too simplistic or outdated and require a more differentiated introduction. Since the introduction is rather superficial, the specific research questions that drive this study are also not well worked out.

Answer: The introduction has been revised as suggested. The changes have been highlighted in yellow in the revised manuscript.

Pg 10, line 42. This sentence should probably be moved to the beginning of the paragraph.

Answer: The manuscript has been revised accordingly.

Reviewer #2 (Remarks to the Author):

In this manuscript, the authors claimed that nanoclustering of cell surface transmembrane receptor kinase 1 drives the jigsaw-puzzle pattern of plant leaf epidermal cells. The authors focused on the ordered lipid in the formation of polarity in a single cell and examined the functions of the related molecules using live imaging techniques. They also formulated a mathematical model of the polarity formation and provided some experimental evidence to support their claim. While the experimental part seems to be a substantial contribution to this field, mathematical modeling does not represent the actual situation (although carefully written) and does not add significant insight into this work. This problem comes from miscommunication between experimental biologists and mathematicians in the group. The authors should rewrite the mathematical model from scratch or completely remove the mathematical modeling part.

* Their model only deals with the dynamics of molecules on the single plasma membrane, which does not represent the actual experimental situation. In the plant leaf, two plasma membranes sandwich the cell wall of constant thickness, and the indentation region in a cell is the protrusion region in the neighboring cell. The model ignored this aspect. The model should incorporate this aspect, or the experimental biologists in the authors should show experimental results in which isolated single epidermal cells can generate indentation under some condition.

Answer: We thank the reviewer for the reviewer for the positive comments and constructive suggestions. The comment "mathematical modeling does not represent the actual situation" was likely prompted by insufficient description of our motivation for our modeling work and misleading presentation of our models in our original submission. The formation of the puzzle-piece shape in pavement cells involves several key processes, including 1) the establishment of the multi-polarity sites that determine the position and the number of lobes and indentation, 2) the coordination between the complementary lobes and indentation sites to initiate the formation of lobes and indentations, and 3) the expansion at the shoulder regions to increase the depth of lobes and indentations. Our motivation for the modeling work is to understand how auxin promotes the establishment of the initial multi-polarity--the first process. We did not intend to model the actual final shape of the pavement cells. The second process involves the activation of both ROP2 and ROP6 pathways by auxin and their intracellular and cross-cell interactions and the third process likely involves mechanical feedback regulation. These two processes are not addressed in our current study. The current study focuses on how diffusible signal like auxin can initiate the polarity establishment at the plasma membrane (PM). In the model, we simulated the process underlying the auxin-induced generation of laterally segregated polarized domains at the plasma membrane. The significance of our modeling work lies in understanding how a uniform diffusible signal like auxin can generate cell polarity. This type of polarization occurs in several cell types such as root hairs independent of cell-cell interactions. Thus, the fact that we did not model the actual cell shape does not take away the significance of our modeling work, which led to the development of our conceptual framework of linking auxin-induced nanoclustering with cell polarity establishment. By analogy, models developed by others also only capture some aspects of pavement cell morphogenesis, but also are highly significant. For example, mechanical models, such as the models presented by Sapala et al. (2018)¹⁵ and Bidhendi et al. (2019)¹⁶, explain the importance of mechanical stresses and the cell wall components in shaping pavement cells, but do not explain how the asymmetric distribution of cell wall stiffness or components is established and how auxin promotes pavement cell morphogenesis. Our model presentation was misleading by introducing the cell shape change into the model in our previous submission. The shape change was arbitrarily included in our model to display the number of cell polarization sites more clearly, but it was not meant to explain how polarity or microtubules impact cell shapes. However, this also misled the reviewer to think that we are modelling the jigsaw-puzzle pattern formation. To avoid the confusion, we have revised the model description and removed the cell shape change in the model.

* There are at least three previous works that deal with spontaneous pattern formation of plant leaf epidermal cells (Higaki et al., 2017, 2016; Sapala et al., 2018). Two of them utilize buckling instability of the cell wall, and one utilizes interface instability of band-like solution in a reaction-diffusion system, which implements production and degradation of cell wall controlled

by diffusible molecules. The references to these works are missing. The authors should discuss the relationship between these previous models and their model.

Answer: We thank reviewer for the suggestions. As discussed above, the main difference between our model and previously reported models is that our model focuses on the mechanisms underlying the polarity establishment at the PM, specifically, how the polarized membrane domains are generated prior to the cell shape change, while others simulate the whole process of pattern formation. The pattern formation is a complex cellular process requiring elaborate spatiotemporal regulation of components, not only in the PM but also in cell walls. The involvement of cell wall dynamics on the polarity establishment and the interplay between the plasma membrane and the cell wall for shape formation was beyond the scope of this manuscript but our next step to study. This discussion has been added into the revised manuscript. Also, the suggested references have been cited in the revised manuscript. Please see Page 11, lines 26-29.

* The patterns their model generated are not similar to the actual pattern (Fig. 6D, E). The actual patterns consist of smooth curves and seem to have some characteristic length, while their simulation results seem to be random clefts. I want to confirm whether the experimental biologists in the author group are satisfied with this result.

Answer: As mentioned, our model focuses on process of cell polarity establishment, which occurs before visible cell polar growth. In the revised manuscript, the cell shape change was removed from the model and the signaling network was investigated on a fixed cell shape.

* Mathematical analysis and insight of the dynamics of their "mathematical" model are missing. There exists a standard mathematical analysis of why pattern formation takes place (linear stability analysis), and that will help to understand what wavenumber should be selected (=size and number of the structures). Their model is a slightly modified reaction-diffusion model, and it should be easy to reduce the model to an analytically manageable form. Mathematicians in the authors should do more mathematics, not dumbing down the ideas to communicate with biologists. Currently, the model only numerically shows that with a specific parameter set, the model behavior looks similar. At first, they should formulate a new model which includes the plasma membranes of two adjacent cells and cell walls, and reduce the model to undertake mathematical analysis to understand the model behavior, and possibly provide a reliable prediction on the manipulation of the pattern formation. Otherwise, just removing the modeling part may be a good option because the experimental part is good enough to stand alone.

Answer: We thank reviewer for the valuable suggestions. Most modeling works applied the linear stability analysis to reaction-diffusion systems with homogeneous diffusion coefficients. As far as we know, very few of them studied the systems with nonhomogeneous diffusion coefficients, except those with functions of spatial variables only as the diffusion coefficients^{17,18}. In our study, the signaling network was modeled by a nonhomogeneous reaction-diffusion system with the diffusion coefficients depending on the network components. On the domain with small diffusion coefficient, the system becomes convection dominant and the pattern can be formed in the steady states. The suggested mathematical analysis is now added

into the revised manuscript. Please see Page 20, Line 19 – Page 21, Line 17. The linear stability analysis performed in the revision suggests that the initialization of the cell polarity involving auxin, sterols and TMK is achieved due to the reduced diffusion, which is different from the Turing instability. We think the mathematical model is an essential part of our manuscript because it is technically impossible to visualize the entire initiation event of cell polarity from auxin-induced nanoclustering of signaling components to the formation of polarized domains at the plasma membrane. By using the modeling, we are able to simulate the whole initiate event and found that the microtubule-mediated diffusion restriction of TMK/sterol nanodomains is required for generating the stable polarized ROP6 domain at the PM (Fig. 6). This feedback regulatory mechanism on the nanoclustering of signaling components is novel and further confirmed by our genetic and biochemical data.

Reviewer #3 (Remarks to the Author):

This manuscript by Pan et al describes how nano-clustering of membrane proteins contributes to cell polarization. Arabidopsis pavement cells form multiple indentations during cell growth. Authors' group previously revealed that auxin signaling promotes formation of the indentations through the TMK1 receptor kinase, ROP6 GTPase, and microtubules. In the present study, authors show that TMK1 and ordered sterol lipids are clustered upon auxin signaling, which in turn promotes ROP6 activation and their clustering. Furthermore, by using mathematical models, authors predicted that additional feedbacks from microtubules, inhibiting diffusion of TMK1 and of ordered sterol lipids are required for the nano-clustering-induced cell polarization. Finally, authors show that microtubules indeed influence the clustering of TMK1 and ordered lipid sterols. Generally, the research is well organized, data are of high quality, and discussion is thoughtful. This elegant study provides novel insights into the role of nano-clustering for cell polarization, significantly updating our understanding of how cell polarization is achieved through regulating the dynamics of plasma membrane proteins. I have several suggestions.

We thank the reviewer for the positive comments!

1. Previously, authors' group revealed that the ROP2/4-actin pathway promotes formation of lobes, acting as a counterpart of the ROP6 pathway. It would be very interesting if authors discuss specificity of nanoclustering for ROP6 and others ROPs. This may provide readers with useful information to grasp the overview of pavement cell morphogenesis.

Answer: We thank the reviewer for the suggestion. The discussion regarding the specificity of nanoclustering for different ROPs has been added in the revised manuscript. Please see Page 12, Lines 3-11.

2. Authors observed mEGFP-ROP6 and TMK1-GFP expressed in wild type plants, so it is unclear whether these fusion proteins are functional. I suggest authors mention that these (or similar) constructs complemented mutants in previous studies.

Answer: We thank the reviewer for the suggestion. The relevant information has been added in the revised manuscript. Please see Page 13, Lines 28-29 and Page 14, Lines 10-11.

3. I was a bit confused because authors added a feedback from ROP6 to TMK1 and lipids after showing co-localization of CA-ROP6 and microtubules (Fig S6). I suppose that authors conceived microtubules trap TMK1 and lipids, which in turn promotes clustering of active ROP6, resulting in the accumulation of active ROP6 along microtubules. It would be better to add some explanation to connect the result and hypothesis. In this case, however, I think it is also possible that the microtubules restrict the diffusion of active ROP6 directly or indirectly via RIC1 or other microtubule-associated proteins, then active ROP6 does promote clustering of TMK1 and sterols along microtubules. Could authors discuss or examine this point?

Answer: We thank the reviewer for these valuable comments. To exam how microtubules affect the dynamics of sterol-rich nanodomains, we further studied the spatial relation between flotillin1 and microtubules in pavement cells co-expressing flotillin1::flotillin1-mVenus and UBQ::mScarlet-MAP4. As shown in Supplementary Figure 9C, the majority of the flotillin1 nanoclusters do not associate with cortical microtubules (CMTs) but rather localize within microtubule corrals. Based on these results, we further elaborated our model as follows: First, TMK1/sterol nanodomains provides the platform for converting inactive ROP6 into active ROP6. Then active ROP6 associates with and promotes the CMT ordering. The association of ROP6 proteins with the CMTs makes them leaves the sterol-rich nanodomains, presumably because of steric hindrance effects between CMTs and sterol-rich nanodomains. The CMTs in turn stabilize TMK1/sterols platforms for ROP6 activation. Please see the added discussion in Page 11, Lines 7-15.

To clarify the relationship between ROP6, microtubules, and TMK1, it may be useful if authors examine whether CArop6 expression affects TMK1/flotillin1 dynamics in the presence of oryzalin. Also, I suggest that authors observe co-localization of ROP6 and TMK1/ flotillin1 to confirm their association in the lipid/protein clusters.

Answer: We thank the review for these suggestions. As suggested, we studied the effect of CArop6 on TMK1 dynamics. As shown in Supplementary Figure 11D, the expression of CArop6 led to increased TMK1 particle size with reduced diffusion rate. This effect was compromised in the presence of oryzalin, suggesting that the intact microtubules is required for the CArop6-induced dynamic changes of TMK1 particles. In addition, we tested the colocalization of ROP6 and flotillin1 in cotyledons co-expressing mEGFP-ROP6 and flotillin1-tagRFP (Supplementary Figure 7). However, the flotillin1 promoter is very weak and flotillin1::flotillin1-tagRFP showed very weak fluorescent signals. In the mock treatment, flotillin1 puncta cannot be distinguished from the background. Therefore, we used the IAA treatment (1 μ m IAA, 10 mins) to promote the formation of larger flotillin1 nanoclusters with enhanced fluorescent intensity. As shown in the Supplementary Figure 7, ROP6 and flotillin1 at least partially reside in the same nanodomains.

4. To confirm the feedback from microtubules to TMK1 and flotillin1, authors analyzed TMK1 and flotillin1 dynamics upon treatment with oryzalin. Although this nicely demonstrated that microtubules indeed influence the clustering of TMK1 and flotillin, the experimental condition seems not to reflect the exact downstream events of ROP6 where ROP6 promotes microtubule ordering through RIC1 and katanin activities. It would be interesting if authors examine dynamics of TMK1 and flotillin1 in ric1 or katanin mutants or overexpression lines. I think this

makes the story more robust. Alternatively, authors may test if taxol promotes microtubule ordering and influences dynamics of TMK1 and flotillin1 in young pavement cells.

Answer: We thank reviewer for this comment. As suggested, we studied the dynamics of TMK1/flotillin1 in *ktn1* mutant. As shown in Supplementary Figures 11A-11B, consistent with the oryzalin treatment, the particles of TMK1 and flotillin1 displayed smaller size and faster diffusion rate in the *ktn1* mutant than in the wild-type mutant. In addition, we further studied the dynamics of TMK1 in the *mor1-1* mutant background. As shown in Supplementary Figure 11C, the relative change of diffusion rate of TMK1 particles after incubating the cotyledons at 29°C for 2hrs is much higher in the *mor1-1* mutant background than in the Columbia WT background.

5. The *tmk1tmk4* mutant exhibits severer developmental phenotype. It would be possible that the phenotypes of ROP6 and flotillin1 in *tmk1tmk4* mutant includes secondary effects of the developmental defects. To eliminate this possibility, I suggest authors test transient treatment with auxin transport inhibitor NPA.

Answer: We thank the reviewer for this suggestion. The effect of NPA treatment on pavement cell is complicated due to the presence of different types of PIN proteins. Therefore, it is difficult to determine the proper timing and concentration of NPA treatment. In addition to PIN1 localized to lobe sites as we described previously, our unpublished work shows that another PIN protein is polarized to the apex of cotyledon margin cells to transport auxin to the tip of cotyledons and the tip-localized auxin provides a global signal for the coordination of cell morphogenesis throughout the entire cotyledon. Due to these complications, we did not perform the NPA treatments.

Minor

Page 6, line 34 “Jaillais’s group showed that~”

Here, another phrase may be preferred such as “Platre et al. (2019) showed that” or “it was shown that” or others.

Answer: We thank the review for the suggestion. The manuscript has been revised.

Page 8, line 26

It is better to remove “a high concentration of”.

Answer: We thank the review for the suggestion. The manuscript has been revised accordingly.

Page 8, line 26

Remove a space from “flotillin 1”.

Answer: We thank the review for the suggestion. The manuscript has been revised accordingly.

References

1. Boutté, Y., Men, S. & Grebe, M. Fluorescent in situ visualization of sterols in Arabidopsis roots. *Nat. Protoc.* (2011). doi:10.1038/nprot.2011.323
2. Ovecka, M. *et al.* Structural sterols are involved in both the initiation and tip growth of root hairs in Arabidopsis thaliana. *Plant Cell* **22**, 2999–3019 (2010).
3. Ge, L. *et al.* Flotillins play an essential role in Niemann-Pick C1-like 1-mediated cholesterol uptake. *Proc Natl Acad Sci U S A* **108**, 551–556 (2011).
4. Kokubo, H. *et al.* Ultrastructural localization of flotillin-1 to cholesterol-rich membrane microdomains, rafts, in rat brain tissue. *Brain Res.* (2003). doi:10.1016/S0006-8993(02)04140-9
5. Rai, A. *et al.* Dynein clusters into lipid microdomains on phagosomes to drive rapid transport toward lysosomes. *Cell* **164**, 722–734 (2016).
6. Fu, Y., Xu, T., Zhu, L., Wen, M. & Yang, Z. A ROP GTPase signaling pathway controls cortical microtubule ordering and cell expansion in Arabidopsis. *Curr Biol* **19**, 1827–1832 (2009).
7. Li, H. *et al.* Phosphorylation switch modulates the interdigitated pattern of PIN1 localization and cell expansion in Arabidopsis leaf epidermis. *Cell Res.* (2011). doi:10.1038/cr.2011.49
8. Platre, M. P. *et al.* Developmental control of plant Rho GTPase nano-organization by the lipid phosphatidylserine. *Science (80-.)*. **364**, 57–62 (2019).
9. Wan, Y. *et al.* Variable-angle total internal reflection fluorescence microscopy of intact cells of Arabidopsis thaliana. *Plant Methods* **7**, 27 (2011).
10. Grossmann, G. *et al.* Green light for quantitative live-cell imaging in plants. *Journal of Cell Science* (2018). doi:10.1242/jcs.209270
11. Martiniere, A. *et al.* Cell wall constrains lateral diffusion of plant plasma-membrane proteins. *Proc. Natl. Acad. Sci. U. S. A.* **109**, 12805–12810 (2012).
12. McKenna, J. F. *et al.* The cell wall regulates dynamics and size of plasma-membrane nanodomains in Arabidopsis. *Proc. Natl. Acad. Sci. U. S. A.* (2019). doi:10.1073/pnas.1819077116
13. Dunn, K. W., Kamocka, M. M. & McDonald, J. H. A practical guide to evaluating colocalization in biological microscopy. *American Journal of Physiology - Cell Physiology* (2011). doi:10.1152/ajpcell.00462.2010
14. Poraty-Gavra, L. *et al.* The Arabidopsis Rho of plants GTPase AtROP6 functions in developmental and pathogen response pathways. *Plant Physiol* **161**, 1172–1188 (2013).
15. Sapala, A. *et al.* Why plants make puzzle cells, and how their shape emerges. *Elife* (2018). doi:10.7554/eLife.32794
16. Bidhendi, A. J., Altartouri, B., Gosselin, F. P. & Geitmann, A. Mechanical Stress Initiates and Sustains the Morphogenesis of Wavy Leaf Epidermal Cells. *Cell Rep.* (2019). doi:10.1016/j.celrep.2019.07.006
17. Benson, D. L., Sherratt, J. A. & Maini, P. K. Diffusion driven instability in an inhomogeneous domain. *Bull. Math. Biol.* (1993). doi:10.1007/BF02460888
18. Benson, D. L., Maini, P. K. & Sherratt, J. A. Unravelling the Turing bifurcation using spatially varying diffusion coefficients. *J. Math. Biol.* (1998). doi:10.1007/s002850050135

The authors reinforced the explanations on the modeling part, which helps to understand the core dynamics of their model. As a result, it becomes more unclear whether their model captures the real dynamics. Also, some questions are not answered. I still think the biological content alone is enough for this paper, and mathematical analysis can be done somewhere else unless their model becomes considerably reasonable in the subsequent modification.

1. Concerning the observation corresponding stage of development, I understand the limitation of the experiment.

2. The reduced model in the main text (page 6, line 6)

$$\begin{cases} u_t = \partial_x \left(\frac{D_1}{1 + (v/k_{vu})^{n_1}} u_x \right) - d_1 u \\ v_t = \partial_x \frac{r}{1 + (u/k_{uv})^{-n_2}} - d_2 v \end{cases}$$

is different from the model described in the reply letter.

$$\begin{cases} u_t = \partial_x \left(\frac{D}{1 + (v/k_{vu})^{n_1}} u_x \right) - d_1 u \\ v_t = \frac{r}{1 + (u/k_{uv})^{-n_2}} - d_2 v \end{cases}$$

I guess the latter is correct, and the authors should remove " ∂_x " in v_t equation in the main text.

3. I am not sure this equation can generate phase separation. It seems total amount of u ($\int u dx$) should be monotonously decreasing because the first term conserves the total amount, and the second term represents decay. Also, since the diffusion coefficient of u ($D(v)$) is always positive, local accumulation of u should not happen (We can show that at the peak point u should always decrease). Therefore it seems the dynamics should look like a relaxation process toward a homogeneous steady state in which the amplitude of u fluctuation is continuously decreasing. Please confirm whether this equation is correct. Alternatively, if the authors showed the actual parameter sets for this simulation, I can reproduce and see whether the author's claim is plausible.

I noticed that some results use special color scheme - "domains with high ROP6 activity" is colored in purple, and "domains with high ROP6 activity" are colored in cyan. I could not find a description of the threshold. With this color scheme, one can pretend simple relaxation process as phase separation by selecting specific favorable timepoints. I would suggest the mathematicians in the authors show the concentration profile of the result, not the binarized color images, to see whether the dynamics are part of the relaxation process toward the steady-state or actual phase separation.

As mathematicians in the authors should know, standard Allen-Cahn or Cahn-Hilliard equation can generate phase separation, and they should use formulation, which can be reduced to such previously well-studied equations since the dynamics itself is not very new.

4. Are the results of numerical simulations part of the main text? The format seems to be different from that required by the journal (they should be collected to the Figure).

5. In general, the description of linear stability analysis is difficult to read. There are some points I cannot understand with the current description (I am a frequent user of linear stability analysis, and this is somewhat strange).

model or not? (I could not find numerical result of 2-species model). The authors described "If $D_1(v)$ and $D_2(u)$ are close to zero, the system becomes convection dominant and exhibits hyperbolic property", but this requires explanation. Since we can spatially rescale the system, the absolute value of diffusion coefficients should not affect the instability (changing D_u and D_v to D_u and D_v by some scaling factor ϵ can change the size of the pattern but not the instability itself).

This part is not answered. "convection dominant" "hyperbolic property" are difficult to interpret. Please simply show why the eigenvector is positive in some k in the above equation.

Additional point: "Therefore, with the diffusion rate significantly larger than zero,

the homogeneous steady state is also stable": dynamics of homogeneous steady-state should be determined by reaction term only (in other words $k = 0$) and should not be affected by diffusion coefficient.

6. Variables in the main text should be italicized.

Responses to referees

Reviewers' comments:

Reviewer #1 (Remarks to the Author):

I appreciate the authors' efforts to improve their manuscript and satisfactorily address my concerns and comments. I have only a couple minor suggestions:
Blue signal on black background (Supp. Fig 2 + 8) has very poor contrast. Please change the LUTs in those images to greyscale.

We thank the reviewer for the suggestion. As suggested, the LUTs in these images have been changed to greyscale in the revised manuscript.

It is somewhat surprising to see low probability values of $P \leq 0.0001$ being indicated for several diffusion coefficient measurements with highly similar data distributions (e.g. Fig 5, Supp. Figure 8B.) Please verify that no mistake has occurred; please add the number of measurements to the legends.

We thank the reviewer for the suggestion. We have confirmed the correctness of statistical analyses and added the number of measurements in the figure legends.

Again, we highly appreciate the reviewer's time and constructive comments during the reviewing process.

Guido Grossmann

Reviewer #2 (Remarks to the Author):

After the revision, the author's intention becomes more transparent. However, I still have some concerns in the model part. Considerable modifications should be made in modeling part. My comment is colored red.

I understand the modification. However, in that case, the authors should show the experimental data of the very initial phase of pattern formation, which is comparable to their simulation images (like Suppl. Fig. 8).

Answer: We thank the reviewer for the suggestion. As mentioned in the manuscript, it is technically impossible to image the entire initiation event from nanoclustering to the formation of polarized domains along the anticlinal wall. Analyzing the distribution of protein/lipid nanodomains at Stage I cells in a single time point (as an example of Supplementary Figure 8A) cannot provide reliable quantifications on the number of stabilized polar domains, because the nanodomains in these images have a very dynamic assembly-disassembly behavior

(Supplementary Video 3) and we cannot differentiate the stabilized polar domains from the transient nanodomains in the snap shot images. In contrast, the number of polarization sites in stage II cells are firmly established and can be served as an indicator of stabilized polar domains as these sites are the outcomes of the dynamic polarized domains generated in Stage I cells. Therefore, images of stage II cells serve as a better experimental data in terms of reliable quantitative analyses of the # of polarized domains generated from the initiation events. In addition, it is technically difficult to separate the plasma membrane regions between the complementary lobe and indentations in Stage I cells. For these reasons, we compared images of Stage II cells with the *in silico* simulations. In our opinion, the inability to visualize the initiation events in stage I cells should not be viewed as a drawback in this study. Instead, it highlights the power of modeling, giving specific predictions regarding the roles of cortical microtubules in the auxin-induced polarity establishment, which we experimentally validated and thus lead to new insights into the mechanisms for nanoclustering-based polarity establishment that are otherwise elusive.

I confirmed the description in the Introduction.

I confirmed the change in the simulation part. As described above, the authors should show images of corresponding earlier stages. Otherwise, it is still difficult to correlate simulation results and experimental data.

Answer: Please see the answers above for the first comment.

The modification is unsatisfactory. My main request is to provide a minimal model which

- (a) is in principle the same as full model
- (b) reproduces the dynamics
- (c) is analytically manageable
- (d) helps understanding the mechanism by analysis

Conditions (b) and (d) are not clear to me.

(a) Do the authors insist they could reproduce the pattern using the reduced 2-species model or not? (I could not find numerical result of 2-species model). The authors described "If $D_1(v)$ and $D_2(u)$ are close to zero, the system becomes convection dominant and exhibits hyperbolic property", but this requires explanation. Since we can spatially rescale the system, the absolute value of diffusion coefficients should not affect the instability (changing D_u and D_v to D_u and D_v by some scaling factor s can change the size of the pattern but not the instability itself).

Answer: We thank the reviewer for the excellent suggestions. As suggested, we performed the numerical simulations on the reduced model to verify that similar patterns can also be generated using the reduced model in wild type (Please see Page 20 Line 25 – Page 23 Line 9). Some perturbation conditions were also tested for this reduced model and similar results were obtained qualitatively when compared with the wild type condition. Notice that the reduced model only included the most important components to illustrate the mechanism, therefore only some

perturbations could be studied on this model as shown in the following. Specifically, the reduced model is formulated as below:

$$\begin{cases} u_t = \partial_x \left(\frac{D}{1 + (v/k_{vu})^{n_1}} u_x \right) - d_1 u \\ v_t = \frac{r}{1 + (u/k_{uv})^{-n_2}} - d_2 v \end{cases}$$

where u represents a certain molecule on the plasma membrane with a noisy homogeneous distribution initially, v represents another component which is promoted by u and reduces the diffusion of u as a feedback. In the wild type simulation shown as below, u can concentrate on several locations, leading to pattern formation.

In the full model, we studied the role of MT by carrying out a simulation for the model without MT. To mimic this study in the reduced model, we removed the component v and the simulation shows u reaches a homogeneous distribution, i.e., no pattern can be formed, as shown below.

It is impossible to study the TMK mutant in the reduced model, unless a different distribution is applied as the initial condition. Auxin treatment can be studied in the reduced model by perturbing the mean concentration of u in the initial condition. When the initial mean value is increased, u is able to concentrate into larger polarized domains, which is similar to the behavior of high auxin treatment generated from the full model (Supplementary Figure 10). For reduced initial mean value, u localizes into smaller polarized domains, similar to low auxin treatment as shown in Supplementary Figure 10.

So the two-component model is sufficient to give rise to different patterns under different conditions and it suggests that the critical criteria to generate a pattern is a feedback between the concentration of u and the diffusion restrictor v . The molecule u promotes v in a concentration-dependent manner and the stabilized v in turn positively regulates the concentration of u by reducing its diffusion. Such pattern formation is not due to instability as observed in classic Turing patterns. One difficulty for the mathematical analysis of this reduced model is the spatially inhomogeneous diffusion coefficient, i.e., the diffusion rate varies dramatically in space at different scales, therefore, scaling the diffusion coefficient uniformly in space can't be applied in this case.

(d) I could not judge what the underlying mechanism by the current explanation of the minimal model. Is this in principle phase separation or instability with very wide unstable range, or something else? A system with a wide unstable range like the Keller-Segel model can generate a seemingly more random pattern. Or, if this system uses phase separation in principle, then the standard TDGL equation may be a better option for the minimal model (we could easily mimic the pattern itself with one variable model). The authors should describe what is the principle mechanism of the pattern formation is from reduced model.

Answer: We apologize for the confusion. The principle mechanism of the pattern formation in this work fits well with the theory of lateral phase separation in biological membrane. Nanoclustering of u molecules, represented as high concentration of u in the reduced model, leads to the formation of ordered PM domains which is essential for the organization of v . The stabilized v in turn facilitates nanoclustering of u and the formation of ordered domains by restricting the diffusion of u . This positive feedback through the diffusion process between u and v results in the formation of lateral segregated ordered domains at which u molecules have slower diffusion rates than in the surrounding less-ordered regions. The slower diffusion at the ordered domains eventually leads to the pattern formation.

I do use and understand the importance of mathematical modeling, therefore I want the authors to use it properly. The "mathematical" model should provide some insight into pattern formation mechanism with an analytical method, which is different from the

"computational" model, which only provides superficial similarity. The authors should properly interpret above description of the mechanism (perhaps by the biologist in the group) in mathematical terms in the main text.

Answer: We understand that the model studied in this manuscript is indeed a computational model. Such computational approach is helpful when it is challenging to obtain the corresponding experimental data to verify the hypothetical mechanism. In this work, it is technically impossible to directly image the entire initiation event from auxin-induced nanoclustering of TMK1/sterols to the formation of laterally segregated polar domains. So, we used *in silico* simulations of the biological network involving experimental identified interactions and calibrated parameter values to test the hypothesis. In detail, the diffusion coefficients in the model were chosen to be at the same order as the experimental quantification and the initial conditions on TMK and Lipid were chosen to satisfy the same ratio as the experimental quantification. We also calibrated the model by using the data of auxin treatment, which makes the simulation results more biologically relevant. We agree that mathematical models with rigorous analyses are beautiful tools to understand the underlying mechanism. However, due to the nonlinear and spatially nonhomogeneous diffusion rates, it is challenging to perform mathematical analysis on this model. The analyses suggested by the reviewer have been carried out for a reduced model to some extent, which provides more insights on the underlying mechanism of pattern formation. We have also added the description of the mechanism in the Methods section (Please see Page 22 Line28 to Page 23 Line 9) along with the new numerical data of the reduced model.

Reviewer #3 (Remarks to the Author):

I highly evaluate the authors to have conducted several additional experiments. The results support their conclusion. I would just like to point some typos in figure legends.

Supplementary Fig7, 1 um should be 1uM.

We thank the reviewer for pointing this out. This has been corrected in the revised manuscript.

Supplementary Fig9, YFP-CAropt6 should be YFP-CArop6. MAP4-mScarlet should be mScarlet-MAP4. Add or remove a space to unify the format of scale bar (3 um or 3um).

We thank the reviewer for pointing this out. This has been corrected in the revised manuscript.

Again, we appreciate the reviewer's valuable comments and suggestions during the reviewing process.

Reviewer #1 (Remarks to the Author):

I appreciate the authors' efforts to improve their manuscript and satisfactorily address my concerns and comments. I have only a couple minor suggestions:

Blue signal on black background (Supp. Fig 2 + 8) has very poor contrast. Please change the LUTs in those images to greyscale.

It is somewhat surprising to see low probability values of $P \leq 0.0001$ being indicated for several diffusion coefficient measurements with highly similar data distributions (e.g. Fig 5, Supp. Figure 8B.) Please verify that no mistake has occurred; please add the number of measurements to the legends.

Guido Grossmann

Reviewer #2 (Remarks to the Author):

After the revision, the author's intention becomes more transparent. However, I still have some concerns in the model part. Considerable modifications should be made in modeling part. My comment is colored red.

* Their model only deals with the dynamics of molecules on the single plasma membrane, which does not represent the actual experimental situation. In the plant leaf, two plasma membranes sandwich the cell wall of constant thickness, and the indentation region in a cell is the protrusion region in the neighboring cell. The model ignored this aspect. The model should incorporate this aspect, or the experimental biologists in the authors should show experimental results in which isolated single epidermal cells can generate indentation under some condition.

Answer: We thank the reviewer for the reviewer for the positive comments and constructive suggestions. The comment "mathematical modeling does not represent the actual situation" was likely prompted by insufficient description of our motivation for our modeling work and misleading presentation of our models in our original submission. The formation of the puzzle- piece shape in pavement cells involves several key processes, including 1) the establishment of the multi-polarity sites that

determine the position and the number of lobes and indentation, 2) the coordination between the complementary lobes and indentation sites to initiate the formation of lobes and indentations, and 3) the expansion at the shoulder regions to increase the depth of lobes and indentations. Our motivation for the modeling work is to understand how auxin promotes the establishment of the initial multi-polarity--the first process. We did not intend to model the actual final shape of the pavement cells. The second process involves the activation of both ROP2 and ROP6 pathways by auxin and their intracellular and cross-cell interactions and the third process likely involves mechanical feedback regulation. These two processes are not addressed in our current study. The current study focuses on how diffusible signal like auxin can initiate the polarity establishment at the plasma membrane (PM). In the model, we simulated the process underlying the auxin-induced generation of laterally segregated polarized domains at the plasma membrane. The significance of our modeling work lies in understanding how a uniform diffusible signal like auxin can generate cell polarity.

This type of polarization occurs in several cell types such as root hairs independent of cell-cell interactions. Thus, the fact that we did not model the actual cell shape does not take away the significance of our modeling work, which led to the development of our conceptual framework of linking auxin-induced nanoclustering with cell polarity establishment. By analogy, models developed by others also only capture some aspects of pavement cell morphogenesis, but also are highly significant. For example, mechanical models, such as the models presented by Sapala et al. (2018)¹⁵ and Bidhendi et al. 16 (2019) pavement cells, but do not explain how the asymmetric distribution of cell wall stiffness or components is established and how auxin promotes pavement cell morphogenesis. Our model presentation was misleading by introducing the cell shape change into the model in our previous submission. The shape change was arbitrarily included in our model to display the number of cell polarization sites more clearly, but it was not meant to explain how polarity or microtubules impact cell shapes. However, this also misled the reviewer to think that we are modelling the jigsaw-puzzle pattern formation. To avoid the confusion, we have revised the model description and removed the cell shape change in the model.

I understand the modification. However, in that case, the authors should show the experimental data of the very initial phase of pattern formation, which is comparable to their simulation images (like Suppl. Fig. 8).

* There are at least three previous works that deal with spontaneous pattern formation of plant leaf epidermal cells (Higaki et al., 2017, 2016; Sapala et al., 2018). Two of them utilize buckling instability of the cell wall, and one utilizes interface instability of band-like solution in a reaction-diffusion system, which implements production and degradation of cell wall controlled by diffusible molecules. The references to these works are missing. The authors should discuss the relationship between these previous models and their model.

Answer: We thank reviewer for the suggestions. As discussed above, the main difference between our model and previously reported models is that our model focuses

on the mechanisms underlying the polarity establishment at the PM, specifically, how the polarized membrane domains are generated prior to the cell shape change, while others simulate the whole process of pattern formation. The pattern formation is a complex cellular process requiring elaborate spatiotemporal regulation of components, not only in the PM but also in cell walls. The involvement of cell wall dynamics on the polarity establishment and the interplay between the plasma membrane and the cell wall for shape formation was beyond the scope of this manuscript but our next step to study. This discussion has been added into the revised manuscript. Also, the suggested references have been cited in the revised manuscript. Please see Page 11, lines 26-29.

I confirmed the description in the Introduction.

* The patterns their model generated are not similar to the actual pattern (Fig. 6D, E). The actual patterns consist of smooth curves and seem to have some characteristic length, while their simulation results seem to be random clefts. I want to confirm whether the experimental biologists in the author group are satisfied with this result.

Answer: As mentioned, our model focuses on process of cell polarity establishment, which occurs before visible cell polar growth. In the revised manuscript, the cell shape change was removed from the model and the signaling network was investigated on a fixed cell shape.

I confirmed the change in the simulation part. As described above, the authors should show images of corresponding earlier stages. Otherwise, it is still difficult to correlate simulation results and experimental data.

* Mathematical analysis and insight of the dynamics of their "mathematical" model are missing. There exists a standard mathematical analysis of why pattern formation takes place (linear stability analysis), and that will help to understand what wavenumber should be selected (=size and number of the structures). Their model is a slightly modified reaction-diffusion model, and it should be easy to reduce the model to an analytically manageable form. Mathematicians in the authors should do more mathematics, not dumbing down the ideas to communicate with biologists. Currently,

the model only numerically shows that with a specific parameter set, the model behavior looks similar. At first, they should formulate a new model which includes the plasma membranes of two adjacent cells and cell walls, and reduce the model to undertake mathematical analysis to understand the model behavior, and possibly provide a reliable prediction on the manipulation of the pattern formation. Otherwise, just removing the modeling part may be a good option because the experimental part is good enough to stand alone.

Answer: We thank reviewer for the valuable suggestions. Most modeling works applied the linear stability analysis to reaction-diffusion systems with homogeneous diffusion coefficients. As far as we know, very few of them studied the systems with nonhomogeneous diffusion coefficients, except those with functions of spatial variables only as the diffusion coefficients reaction-diffusion system with the diffusion coefficients depending on the network components. On the domain with small diffusion coefficient, the system becomes convection dominant and the pattern can be formed in the steady states. The suggested mathematical analysis is now added. In our study, the signaling network was modeled by a nonhomogeneous into the revised manuscript. Please see Page 20, Line 19 – Page 21, Line 17. The linear stability analysis performed in the revision suggests that the initialization of the cell polarity involving auxin, sterols and TMK is achieved due to the reduced diffusion, which is different from the Turing instability.

The modification is unsatisfactory. My main request is to provide a minimal model which

- (a) is in principle the same as full model
- (b) reproduces the dynamics
- (c) is analytically manageable
- (d) helps understanding the mechanism by analysis

Conditions (b) and (d) are not clear to me.

(a) Do the authors insist they could reproduce the pattern using the reduced 2-species model or not? (I could not find numerical result of 2-species model). The authors described "If $D_1(v)$ and $D_2(u)$ are close to zero, the system becomes convection

dominant and exhibits hyperbolic property", but this requires explanation. Since we can spatially rescale the system, the absolute value of diffusion coefficients should not affect the instability (changing D_u and D_v to $s D_u$ and $s D_v$ by some scaling factor s can change the size of the pattern but not the instability itself).

(d) I could not judge what the underlying mechanism by the current explanation of the minimal model. Is this in principle phase separation or instability with very wide unstable range, or something else? A system with a wide unstable range like the Keller-Segel model can generate a seemingly more random pattern. Or, if this system uses phase separation in principle, then the standard TDGL equation may be a better option for the minimal model (we could easily mimic the pattern itself with one variable model). The authors should describe what is the principle mechanism of the pattern formation is from reduced model.

We think the mathematical model is an essential part of our manuscript because it is technically impossible to visualize the entire initiation event of cell polarity from auxin-induced nanoclustering of signaling components to the formation of polarized domains at the plasma membrane. By using the modeling, we are able to simulate the whole initiate event and found that the microtubule-mediated diffusion restriction of TMK/sterol nanodomains is required for generating the stable polarized ROP6 domain at the PM (Fig. 6). This feedback regulatory mechanism on the nanoclustering of signaling components is novel and further confirmed by our genetic and biochemical data.

I do use and understand the importance of mathematical modeling, therefore I want the authors to use it properly. The "mathematical" model should provide some insight into pattern formation mechanism with an analytical method, which is different from the "computational" model, which only provides superficial similarity. The authors should properly interpret above description of the mechanism (perhaps by the biologist in the group) in mathematical terms in the main text.

Reviewer #3 (Remarks to the Author):

I highly evaluate the authors to have conducted several additional experiments. The results support their conclusion. I would just like to point some typos in figure legends.

Supplementary Fig7, 1 um should be 1uM.

Supplementary Fig9, YFP-CAropt6 should be YFP-CArop6. MAP4-mScarlet should be mScarlet-MAP4.

Add or remove a space to unify the format of scale bar (3 um or 3um).

Responses to Referees

The authors reinforced the explanations on the modeling part, which helps to understand the core dynamics of their model. As a result, it becomes more unclear whether their model captures the real dynamics. Also, some questions are not answered. I still think the biological content alone is enough for this paper, and mathematical analysis can be done somewhere else unless their model becomes considerably reasonable in the subsequent modification.

Response: We agree with the reviewer that our modeling needs further modifications, which will be time consuming. As pointed out by this reviewer, the experimental data are sufficient for the publication of our work in NC, and thus in the revised version of our manuscript, the modeling part is deleted.

1. Concerning the observation corresponding stage of development, I understand the limitation of the experiment.

2. The reduced model in the main text (page 6, line 6)

$$\begin{cases} u_t = \partial_x \left(\frac{D_1}{1 + (v/k_{vu})^{n_1}} u_x \right) - d_1 u \\ v_t = \partial_x \frac{r}{1 + (u/k_{uv})^{-n_2}} - d_2 v \end{cases}$$

is different from the model described in the reply letter.

$$\begin{cases} u_t = \partial_x \left(\frac{D}{1 + (v/k_{vu})^{n_1}} u_x \right) - d_1 u \\ v_t = \frac{r}{1 + (u/k_{uv})^{-n_2}} - d_2 v \end{cases}$$

I guess the latter is correct, and the authors should remove " ∂_x " in v_t equation in the main text.

3. I am not sure this equation can generate phase separation. It seems total amount of u ($\int u dx$) should be monotonously decreasing because the first term conserves the total amount, and the second term represents decay. Also, since the diffusion coefficient of u ($D(v)$) is always positive, local accumulation of u should not happen (We can show that at the peak point u should always decrease). Therefore it seems the dynamics should look like a relaxation process toward a homogeneous steady state in which the amplitude of u fluctuation is continuously decreasing. Please confirm whether this equation is correct. Alternatively, if the authors showed the actual parameter sets for this simulation, I can reproduce and see whether the author's claim is plausible.

I noticed that some results use special color scheme - "domains with high ROP6 activity" is colored in purple, and "domains with high ROP6 activity" are colored in cyan. I could not find a description of the threshold. With this color scheme, one can pretend simple relaxation process as phase separation by selecting specific favorable timepoints. I would suggest the mathematicians in the authors show the concentration profile of the result, not the binarized color images, to see whether the dynamics are part of the relaxation process toward the steady-state or actual phase separation.

As mathematicians in the authors should know, standard Allen-Cahn or Cahn-Hilliard equation can generate phase separation, and they should use formulation, which can be reduced to such previously well-studied equations since the dynamics itself is not very new.

4. Are the results of numerical simulations part of the main text? The format seems to be different from that required by the journal (they should be collected to the Figure).

5. In general, the description of linear stability analysis is difficult to read. There are some points I cannot understand with the current description (I am a frequent user of linear stability analysis, and this is somewhat strange).

model or not? (I could not find numerical result of 2-species model). The authors described "If $D_1(v)$ and $D_2(u)$ are close to zero, the system becomes convection dominant and exhibits hyperbolic property", but this requires explanation. Since we can spatially rescale the system, the absolute value of diffusion coefficients should not affect the instability (changing D_u and D_v to D_u and D_v by some scaling factor s can change the size of the pattern but not the instability itself).

This part is not answered. "convection dominant" "hyperbolic property" are difficult to interpret. Please simply show why the eigenvector is positive in some k in the above equation.

Additional point: "Therefore, with the diffusion rate significantly larger than zero, the homogeneous steady state is also stable": dynamics of homogeneous steady-state should be determined by reaction term only (in other words $k = 0$) and should not be affected by diffusion coefficient.

6. Variables in the main text should be italicized.